# Determination and climatology of diurnal cycle of atmospheric mixing layer height over Beijing 2013-2018: Lidar measurements and implication for air pollution

Haofei Wang[1,2], Zhengqiang Li[1*], Yang Lv[1,2], Ying Zhang[1], Hua Xu[1], Jianping Guo[3], Philippe Goloub[4]

1. State Environmental Protection Key Laboratory of Satellite Remote Sensing, Aerospace Information Research Institute, Chinese Academy of Sciences, Beijing, 100101, China

2. University of Chinese Academy of Sciences, Beijing, 100101, China

3. State Key Laboratory of Severe Weather, Chinese Academy of Meteorological Sciences, Beijing, 100081, China

4. Laboratoire d'Optique Atmospherique, UMR8518, CNRS – Universit ´e de Lille 1, Villeneuve d'Ascq, Lille, 59000, France

∗ *Correspondence to:* Zhengqiang Li (Email: lizq@radi.ac.cn)

**Abstract.** The atmospheric mixing layer height (MLH) determines the space where pollutants diffuse and thus is conducive to the estimation of the pollutant concentration near the surface. The study evaluates the capability of lidar to describe the evolution of atmospheric mixing layer and then presents a long term observed climatology of MLH diurnal cycle. Detection of the mixing layer heights ($MLH_L$ and $MLH_L$') using wavelet method based on lidar observations was operated from January 2013 to December 2018 in the Beijing urban area. The two dataset results are compared with radiosonde as case studies and statistical forms. $MLH_L$ shows good performance to calculate the convective layer height at daytime and the residual layer height at night. While $MLH_L$' has the potential to describe the stable layer height as radiosonde at night, the performance is limited due to the high range gate of lidar. A nearly six year climatology for diurnal cycle of MLH is calculated for convective and stable conditions using the dataset of $MLH_L$ from lidar. The maximum $MLH_L$ characteristics of seasonal change in Beijing indicate that it is low in winter ($1.404 \pm 0.751$ km) and autumn ($1.445 \pm 0.837$ km), and high in spring ($1.647 \pm 0.754$ km) and summer ($1.526 \pm 0.581$ km). A significant phenomenon is found that from 2014 to 2018, the magnitude of diurnal cycle of $MLH_L$ increase year by year, with the values of $1.291 \pm 0.646$ km, $1.435 \pm 0.755$ km, $1.577 \pm 0.739$ km, $1.597 \pm 0.701$ km, and $1.629 \pm 0.751$ km, respectively. It may partly benefit from the improvement of air quality. As to converting the column optical depth to the surface pollution, the calculated $PM_{2.5}$ using $MLH_L$ data from lidar shows better accuracy than that from radiosonde, compared with observational $PM_{2.5}$. Additionally, the accuracy of calculated $PM_{2.5}$ using $MLH_L$ shows a diurnal cycle in the daytime, with the peak at time of 14 LST. The study provides a significant dataset of $MLH_L$ based on measurement and could be an effective reference to atmospheric models for surface air pollution calculation and analysis.

## 1. Introduction

The height of mixing layer (MLH) is a crucial parameter for near-surface air quality forecast, pollutants dispersion and quantification of pollutant emissions (Haeffelin et al., 2012; Seibert et al., 2000; Baars et al., 2008; Liu and Liang, 2010; Bruine

et al., 2017). The pollutants discharged in the boundary layer diffuses vertically under the drive of turbulence (Gan et al., 2011; Monks et al., 2009; Guo et al, 2016), and finally becomes completely mixed over this layer if sufficient time is given (Emeis et al., 2008). The MLH determines the space where pollutants diffuse and thus is conducive to the estimation of the pollutant concentration near the surface which might be detrimental to health of human and ecosystems (Emeis et al., 2007; Collaud et al., 2014; Singh et al., 2016; Mues et al., 2017).

Within the planetary boundary layer (PBL) the height of the mixing layer (ML) is defined as the height up to which vertical dispersion by turbulent mixing of air pollutants takes place due to the thermal structure of the PBL (Seibert et al., 2000; Schafer et al., 2006; Emeis et al., 2007).MLH depends largely on the synoptic weather situation (Emeis et al., 2008). MLH can be estimated by the detection of variance of the mechanical turbulence, of the temperature enabling convection or of the substance content in the low troposphere. Singh et al.(2016) investigated the evolution of the Local Boundary Layer in the central Himalayan region, using a radar wind profiler detecting wind components based on signal to-noise ratio profile. Collaud et al. (2014) compared the MLH measurement of microwave radiometer from atmospheric temperature profile with other measurement in Swiss plateau. Mues et al. (2017) used the ceilometer to retrieve the MLH based on aerosol backscatter signal in the Kathmandu Valley. These measurement are based on different atmospheric parameters, different measuring instruments and various analysis algorithms, leading to MLH results obtained by different methods inconsistent (Collaud et al., 2014).

In order to realize from the general definition to practical measurements, it is necessary to consider separately the structure of the convective boundary layer (CBL) and the stable boundary layer (SBL). In the case of fair weather days, the PBL height has a well-defined structure and diurnal cycle (Collaud et al., 2014. See Fig.S1 in the supplementary information). The PBL development under strong convection driven mainly by solar heating is called CBL (Collaud et al., 2014). The nocturnal SBL shows a more complex internal structure, including a stable layer caused by radiative cooling from the ground and gradually merges into a neutral layer called the residual layer (RL) (Stull, 1988; Mahrt et al., 1998; Salmond and McKendry, 2005; Collaud et al., 2014). The RL height is the top of the neutral layer and the beginning of the stable free troposphere. The pollutants discharged from the surface at night are restricted to the SBL, while the pollutants emissions on past day tend to stay in the RL. In addition to the dominance of CBL in the afternoon, the SBL and neutral boundary layer may be formed under certain weather conditions (Stull 1988; Poulos et al. 2002; Medeiros et al. 2005; Zhang et al., 2018)

Since most atmospheric column aerosol particles are usually present in atmosphere below MLH, MLH can be used to convert aerosol optical thickness of the column observed by sunphotometer and satellite to the concentration of near-surface pollutants (Sifakis et al., 1998; Emeis et al., 2007). Particulate can be used as an important indicator of atmospheric layering because their vertical distribution is strongly affected by the thermal structure of the atmosphere (Neff and Coulter, 1986). Provided the vertical aerosol distribution adapts rapidly to the variational thermal-dynamic of the boundary layer, MLH can thus be retrieved from the analysis of this aerosol distribution(Emeis et al., 2008).

By the measurement of profile of aerosol, lidar offers a direct and continuous way to monitor the diurnal cycle of the different layers constituting the PBL (Seibert et al., 2000; De Haij et al., 2006;Emeis et al., 2008;. Liu and Liang 2010; Tang et al, 2016; Su et al, 2017, 2019). With recent upgrades of the hardware, ceilometer, as known as automated low power lidar, or automated lidar and ceilometer (ALC), has been demonstrated be capably to determine the MLH (Wiegner et al., 2006; Geiß et al., 2017; Kotthaus et al., 2017; Mues et al., 2017). Recent studies compared remote sensing measurements (lidar, radar wind profiler, microwave radiometer) with radiosonde (RS) (Wiegner et al., 2006; Milroy et al., 2012; Sawyer and Li, 2013; Cimini et al., 2013; Tang et al, 2016; Singh et al., 2016; Mues et al., 2017; Su et al, 2019), of which convection weather cases has good correlation with differences of 100–300 m, while non-convective weather conditions leads to much larger difference in the MLH estimations if the approaches are supposed to measured different structure of ML such as CBL, SBL or RL (Collaud et al., 2014). The meteorological radiosondes usually acquire the MLH in the morning (08:00 LT) and at night (20:00 LT), when the diurnal cycle of ML combined with stable and convective PBL cannot be well characterized.

In the existing studies, numerical simulations, ground-based remote sensing, or meteorological radiosonde are used to obtain the characterization of MLH during short time periods in Beijing, mainly focusing on heavy pollution event (Yang et al., 2005; Quan et al., 2013; Hu et al., 2014; Zhang et al., 2015), which underscores the scarceness of continuous high-resolution measurement for a long time period. Depending on the measured atmospheric parameters and observational uncertainties, different measurement approaches may reveal different aspects of PBL structure (Seibert et al., 2000; Seidel et al., 2010; Beyrich and Leps, 2012). Thus, it is of great significance to apply consistent algorithm to consistent types of atmospheric structure parameter when comparing MLH from different times.

The main aim of this study, therefore, is aimed to present a long term observed climatology of the MLH diurnal cycle based on lidar observations. For that, the capability of lidar to describe the diurnal evolution of mixing layer height is evaluated first. The data and methods used are described in Section 2. Sect. 3 is the result and discussion, which consist of the comparison of lidar-derived MLH with radiosonde measurements, the climatology of MLH in Beijing and implication for surface pollution retrieval. Then, it is concluded in Sect. 4.

## 2. Data and methods

### 2.1. Site and lidar measurements

Beijing is the capital of China with about 20 000 000 citizens. Beijing city is located on flat terrain in the North China Plain (altitude of 20 m -60 m), with Taihang Mountain in the west and Yanshan Mountain in the north (altitude of 1000 m- 1500 m). Similar to many other metropolitan areas, Beijing suffers from episodes of poor air quality, in particular the fine particulate matter ($PM_{2.5}$). In the study, the observatory (116.379 ° E, 40.005 ° N) in the metropolitan area of Beijing is located on the building roof (59 m a.s.l.) of the Institute of Remote Sensing and Digital Earth, Chinese Academy of Sciences. A micro pulse

lidar (CE370, CIMEL, France) was used to detect the atmospheric aerosol structure at this site. The laser used is frequency-doubled Nd:YAG with pulse repetition frequency 4.7 kHz and energy 8-20 μJ. CE370 operates at wavelength of 532nm with the vertical resolution of 15 m and can detect a long range profile up to 30 km every 1 second. For the enhancement of signal noise ratio, 60 profiles are averaged to restore as one thus with the time resolution of 1 minute. The signal received from lidar is processed by subtraction of atmospheric background first, using the averaged value of the signal received from the height between 22 km and 30 km. There is remnants of the previous signal in the system in the absence of optical signal reception, and these signals is called after-pulse. It is monitored every week, and removed from in the receive signal. Due to the design of the lidar, the received view close to the ground does not completely coincide with transmitted view. There exist a detection blind area of lidar and a geometric overlap factor is used to correct the mismatch of field of view. Then, the correction of range (range corrected signal, RCS) is used to retrieve MLH. The profile of RCS is expressed as f (z), with z the measurement height. Logarithm calculation of RCS (expressed in $In(\beta'_{532})$) is presented in the lidar image (Campbell t al. 2003;Yan et al., 2014; Su et al., 2019). MLH estimation from lidar systems is based on the measurement of the sudden drop in aerosol backscatter at top of the mixing layer (Seibert et al., 2000). The period of lidar measurements is from 2013 to 2018, nearly six years. Except for the lidar data of 2013 mainly existing in winter and spring (the month of 1-4 and 11-12), the measurement of 2014-2018 are all annual continued observations.

**2.2. MLH derived from Lidar**

Wavelet transforms are commonly used in many studies for MLH determination from lidar observations (Cohn and Angevine, 2000; Davis et al., 2000; Brooks, 2003; De Haij et al., 2006; Baars et al. 2008; Su et al., 2019). When it is the maximum value of attenuated backscattering profile convolved with Haar function, the corresponding height is MLH. The equation of wavelet is defined as follows:

$$h\left(\frac{z-b}{a}\right) = \begin{cases} 1, & b - \frac{a}{2} \leq z \leq b; \\ -1, & b < z \leq b + \frac{a}{2} \\ 0, & else \end{cases} \quad (1)$$

Where b is the transformation of the equation, where the equation is cantered, and a is the expansion of the equation. The equation of wavelet covariance transformation $W_f(a, b)$, namely, the convolution of $f(z)$ with wavelet function is defined as follows:

$$W_f(a, b) = \frac{1}{a} \int_{z_b}^{z_t} f(z) h\left(\frac{z-b}{a}\right) dz \quad (2)$$

Where $f(z)$ represents RCS in different height, $z_b$ denotes the lower limit of the height of the profile, and $z_t$ represents the upper limit of the height. A valid $MLH_L$ is detected corresponding to the value b when $W_f(a, b)$ reaches the biggest local maximum with a coherent scale of a (Brooks, 2003; De Haij et al., 2006; Emeis et al., 2008). In this study, the expansion a is selected as 420 m, 435 m, 450 m, respectively, and final $W_f(a, b)$ is calculated from the averaged corresponding values.

Another layer, named MLH$_L$' is detected simultaneously by the first local maximum $W_f(a, b)$ from $z_b$, which is assumed to be smaller than or equal to MLH$_L$ (De Haij et al., 2006; Mues et al., 2017). Actually, every local maximum corresponds an aerosol layer and several internal layers appear making the allocation of a local maximum to an atmospheric feature very difficult (Morille et al., 2008; Geiß et al., 2017; Poltera et al., 2017; Kotthaus et al., 2018). Since the absolute maximum in the

vertical gradient of the lidar profiles is characterized by the rapid degrease in pollutants concentration, the MLH$_L$ can be associated with the CBL height during daytime (Haeffelin et al., 2012; Poltera et al., 2017) and the RL height during nightime (Collaud et al., 2014). However, the interpretation of the first local maximum (MLH') is critical.

To form a diurnal cycle of MLH from these several layers, a geodesic approach was applied to pathfinderTURB (Poltera et al., 2017), while COBOLT (Geiß et al., 2017) uses a time–height-tracking approach with moving windows. Nevertheless, these

method are all based on the selection of the lowest detected aerosol layer. The height of the lowest detected aerosol layer was regarded as the daytime MLH and the nocturnal stable boundary layer, respectively, as reported by Mues et al. (2017) and Kotthaus et al. (2018). Su et al. (2019) developed a DTDS algorithm, started with the lowest point and tracked depending time and stability, but the nocturnal MLH with SBL height is not evaluated. Detection of nocturnal boundary-layer heights, in contrast to the residual layer, is a major challenge (Haeffelin et al., 2012; Lotteraner and Piringer, 2016; de Bruine et al., 2017).

Thus, one of the objective of this study is to investigate the usefulness of MLH$_L$' from CE-370 to capture the SBL height over Beijing.

MLH retrievals are eliminated if a cloud flag is marked when the cloud base is found within 6 km from the surface, and a threshold is selected to distinguish between clouds and aerosol layers. To improve the retrieval, a Gaussian filter is applied to retrievals to smooth the temporal variability, and unrealistic outliers are deleted. Due to the limitation of algorithm and

insufficient lidar overlap, the minimum range of the MLH calculation from CE-370 is on the order of 250 m, which is higher than the order of 50 m of ceilometer. It is due to the optical design of ceilometer using the same lens for the emitter and the receiver optical paths, which suffers low signal noise ratio when providing the lower overlap with the limited power transmitted from the optical design. Detecting significant vertical gradients of attenuated backscatter can be challenging (Eresmaa et al., 2012; Haeffelin et al., 2012).Compared to CE-370, ceilometer usually need to a large scale of temporal and vertical averaged,

in the cost of reduction of retrieval in relatively clean atmospheric conditions (de Bruine et al., 2017; Kotthaus et al., 2018).

### 2.3. MLH from Radiosonde

Radiosonde (RS) measurements are one of most widely used methods, especially in China, to derive SBL height and CBL height due to their ability to characterize the thermodynamic and dynamic states of the boundary layer (Piringer et al., 2007; Seidel et al., 2010; Guo et al., 2019).The meteorological radiosondes are measured at the international standard weather station

(39.484 ° N, 116.282 ° E), which was located nearly 11 km far from lidar station. It includes two categories: conventional observations around the year, which are performed at 0000 UTC (0800 LST) in the morning and at 1200 UTC (2000 LST) in

the evening each day; and intensified observations only in summer, which are operated at 0600 UTC (1400 LT) in the afternoon. The observed meteorological parameters includes atmospheric pressure (P), temperature (T), relative humid (RH), wind speed (WS), wind direction (WD), and so on.

The bulk Richardson number ($Ri_b$) is a dimensionless parameter combining the thermal energy and the vertical wind shear, and is widely used in MLH climatology (Seidel et al., 2012; Collaud et al., 2014). $Ri_b$ is defined as the ratio of turbulence associated with buoyancy to that induced by mechanical shear, which is expressed as

$$Ri_b = \frac{gz(\theta(z) - \theta(z_s))}{\theta_{z_s}(U^2(z) + V^2(z))}$$ (3)

where z is the height ($z > z_s$, subscript 's' denote the surface), θ characters virtual potential temperature, U and V indicates the two horizontal wind velocity components, g presents the Earth gravitational constant. The $MLH_{RS}$ corresponds to the first elevation z with $Ri_b$ greater than a critical threshold taken as 0.25 (Stull, 1988; Seidel et al. 2012; Guo et al., 2016, 2019). In most cases, the exact threshold value has only a small impact on the PBL height due to the large slope of $Ri_b$ in this interval (Collaud et al., 2014).

## 2.4 Air pollution model

The data of MLH is usually combined within the atmospheric model to obtain the surface air pollutant concentration. For example, PM$_{2.5}$ remote sensing(PMRS)model, derived by Zhang and Li (2015), have the ability to calculate the mass concentration of PM$_{2.5}$ above ground. The PMRS method is designed to employ currently available remote sensing parameters, including aerosol optical depth (AOD), fine mode fraction (FMF), planetary boundary layer height (PBL height) and atmospheric relative humidity (RH), to derive PM$_{2.5}$ from instantaneous remote sensing measurements under different pollution levels(Zhang and Li, 2015; Li et al., 2016;Yan et al., 2017). PM$_{2.5}$ is calculated following PMRS model as:

$$PM_{2.5} = \frac{AOD}{PBLH} \cdot \frac{FMF \cdot VE_f(FMF) \cdot \rho_{2.5,dry}}{f_0(RH)},$$

Where AOD indicates aerosol optical depth and FMF represents fine mode fraction; $VE_f$ is the ratio of volume and extinction of fine mode aerosol, which can be calculated from FMF, following as $VE_f(FMF) = 0.2887FMF^2 - 0.4663FMF + 0.356$. The parameter $\rho_{2.5,dry}$ indicates the density of dry PM$_{2.5}$, while $f_0(RH)$ presents the particle hydroscopic growth function, which is $f_0(RH) = (1 - RH/100)^{-1}$. PBL height can be derived from remote sensing and radiosonde measurement.

All the parameter is observed by the instruments employed in the same observatory of lidar. The optical parameters of the column aerosols (AOD and FMF.) are obtained by sky-sun photometer (CE318-DP, CIMEL, France), which is affiliated with the Aerosol RObotic NETwork (AERONET) (Holben et al., 1998; Dubovik, 2000). Measurements are automatically scheduled with direct sun irradiance measurements each of about 15 min and angular sky radiance scanning of about 1 h each (Li et al., 2015; Che et al., 2014; Wang et al., 2019). Atmospheric meteorological data (relative humidity-RH, wind speed-WS, wind direction-WD, etc.) are obtained by automatic meteorological monitoring station (BLJW-4). PM$_{2.5}$ mass concentration is

obtained by PM$_{2.5}$ monitor (BAM-1020, MetOne, USA), which shows good agreement with the measurement of national monitoring network near the observatory. All the data is quality controlled and calculated as one hour averaged and the measurement period is from 2014 to 2018. The MLH obtained both from lidar and radiosonde within the period is used in the model to calculate the surface PM$_{2.5}$. For convenient comparison with air quality and meteorological parameters, all MLH results are one hour averaged.

## 3. Results and Discussions

### 3.1. MLH operational measurement

A selection of typical atmospheric conditions included in the data set of lidar measurement are plotted in Fig. 1 - Fig. 4. The heights of the mixing layer, MLH$_L$ and MLH$_L$', are obtained from different criteria using the wavelet covariance transform method. As shown in Fig.1, the development of a convective mixing layer could clearly be observed, with a sharp decrease in aerosol backscatter between the mixing layer and the free atmosphere. MLH$_{RS}$ is also presented accompany with the evolution of MLH$_L$ and MLH$_L$'.

As Fig.1a shown, MLH$_L$ and MLH$_L$' increase when sunrise. In 20170302, MLH$_L$ and MLH$_L$' shows obvious diurnal cycle, with maximum up to 1.0 km. During the evening of 20170303 and the early morning of 20170304, the aerosol layers presents visually obvious two layers, and MLH$_L$' characterizes the first layer height and MLH$_L$ retrieves the upper layer top. In the next day, due to the existence of cloud, the MLH results is discrete. In the evening and early morning MLH$_L$' deviated from MLH$_L$ and approached to MLH$_{RS}$. As shown as the vertical profile of lidar (RCS, wavelet) and radiosonde (RH, T) at 2000 (LST) of 20170303(Fig.1b), both MLH$_L$' and MLH$_{RS}$ demonstrates 0.53 km, while MLH$_L$ shows 1.22 km. Fig.1c indicates that MLH$_L$' (0.86 km) approaches to MLH$_{RS}$ (0.61 km), albeit a little 0.25 km higher, and much lower than MLH$_L$ (1.62 km). In this cases, the result of MLH$_L$', present by first local maximal aerosol gradient, agree well with MLH$_{RS}$. However, it would not always true, just like Fig.1d. MLH$_L$' (0.752 km) is much higher than MLH$_{RS}$ (0.243 km), but equal to MLH$_L$. It is related to that the stable layer height obtained from radiosonde in the case is out of the range of lidar detection (0.255 km), in which, that is, MLH$_L$' from lidar is disabled to determine the stable layer height.

In the summer time (JJA) when the radiosonde is additionally launched at 1400 LST to detect the convective boundary layer, it can provide the comparison between lidar and RS measurement in the afternoon. As shown in Fig.2a, MLH$_L$ undergoes a rapid increase in the morning and reach the peak in the afternoon, while MLH$_L$' grow with a smaller magnitude. In 20140825 and 20140826, the aerosol load is relative low, and MLH$_L$ reaches the peak around 3.0 km in the afternoon, while a lower MLH$_L$ peak in 20140827 with a high aerosol content. In the afternoon of these three days, MLH$_L$ shows consistent with MLH$_{RS}$, while MLH$_L$' is frequently under MLH$_{RS}$. The measurement of RS in the evening and early morning presents very low value, with the order of 0.2 km – 0.3 km. The detailed information represented in Fig.2b shows that MLH$_L$ is equal to MLH$_{RS}$, which

reaches up to 2.95km, while $MLH_L$' is only 1.24 km. Under clear convective conditions of 20140825 and 20140826, when vertical gradients in the aerosol load, indicated by RCS, is weak, lidar can still catch the good MLH result compared to radiosonde, as shown in Fig.2c and 2d that $MLH_L$ (2.25 km and 2.18 km) approaches to $MLH_{RS}$ (2.96 km and 2.50 km). The a little lower value of $MLH_{RS}$ is associated with that aerosol within the mixing layer needs some time to adjust to the thermal structure, and exist a delay to reach the thermodynamics PBL height (Stull, 1988; Collaud et al., 2014).

As Fig.3a shown that the peaks of the three days gradually increase, with the value of 1.0 km, 2.0 km and 2.5 km. Due to high temporal variability of the distribution of aerosol, $MLH_L$ presents incontinuity in 20170617. $MLH_L$ present good evolution of mixing layer height in 20170615 and 20170617 compared to $MLH_{RS}$. $MLH_L$' corresponds to $MLH_{RS}$ for most time in the morning and evening from 20170615 to 20170617. But when stable layer height from radiosonde is around or below 0.25 km, for example 0800 LST on 20170616 with the RS measurement of 0.27 km (Fig.3b), $MLH_L$' misses the height of SBL, but point to the height of residual layer (0.61 km).When stable layer height is higher than 0.25 km, $MLH_L$' (0.62 km) tend to approach to $MLH_{RS}$ (0.62 km) (Fig.3d.). However, in the afternoon $MLH_L$ is used to be close to $MLH_{RS}$ than $MLH_L$' (Fig.3c.). As Fig.4a shown, $MLH_L$ ($MLH_L$') presents the diurnal cycle with the maximum of 1.2 km, while the next two days stays stable in the whole day, and the height of SBL is missed by $MLH_L$' (Fig. 4b- Fig. 4d).

**3.2. Inter-comparison of different MLH approaches**

A comparison of MLH estimated by lidar and radiosonde of 0800 and 2000 LST, is shown in Figure 5. The same observation period of nearly six year (2013-2018) is considered, of which the data is continued except for 2013. As shown in the histograms of Fig.5, the total column is the annual relative frequency and the different colors indicate the contribution of each season to the total. There is a wide discrepancy between $MLH_L$ and $MLH_{RS}$ at the time of 0800 and 2000 LST. The frequency of MLH from radiosonde lower than 0.25 km is nearly 35%,where it is no data for $MLH_L$ from lidar due to the limited detection range. This lower values mainly occurs in winter and autumn, when it tends to present lower MLH (Tang et al., 2016). Specifically, the rate of $MLH_L$ from lidar smaller than 0.5 km is nearly 18% and 12%, respectively, at 0800 LST and 2000 LST, while the corresponding frequency of radiosonde is beyond 75% and 66%. The frequency of larger $MLH_L$ value at the time of 2000 LST is bigger than that of 0800 LST, both from lidar and radiosonde. It is reasonable that the residual layer have not yet collapsed entirely at 2000 LST, while the CBL have not developed well in the early morning. As for the $MLH_L$', its distribution trend is more similar to $MLH_{RS}$ than $MLH_L$ (See Fig.S2), and that the correlation between $MLH_L$' and $MLH_{RS}$ is a little higher than that between $MLH_L$ and $MLH_{RS}$, in spite that it is still not good (See Fig.S3 and Fig.S4). It indicates that $MLH_L$' have the potential to determine the SBL height as radiosonde does.

As to the seasonal variation of both lidar and RS measurement at 0800 LST, the frequency of larger $MLH_L$ value in summer is minimal, indicating summer MLH is lower than other season. As for radiosonde, $MLH_L$ lower than 0.25 km mostly

distributes in winter, with the rate of around 15% for both 0800 and 2000 LST, and the frequency decreases rapidly when $MLH_L$ gets larger than 0.25 km.

The poor agreement between MLH from lidar and $MLH_{RS}$ is also reported in the study of Su et al. (2019), in which shows that the correlation of PBL height measurement between lidar and radiosonde is 0.14 at 0630 LST. The significant scatter in the morning and evening is associated with complicated structure of boundary layer, as indicated by the existence of stable boundary layer and residual layer (Su et al., 2019; Tang et al., 2016). In this study, no matter $MLH_L$ and $MLH_L$', more than 35% measurement of SBL height is not within the scope of the lidar detection. Additionally, in the evening and early morning of some cases, a sufficiently clear variety cannot be found in the backscatter profile at the top of the SBL, within the previously well-mixed layer (Russell et al., 1974; Seibert et al., 2000).

The comparison of $MLH_L$ and $MLH_{RS}$ at time of 1400 LST in summer is presented in Fig.6, both of them mainly indicating the CBL height. $MLH_L$ shows very good agreement with $MLH_{RS}$, with correlation coefficient of 0.692 and RMSE of 0.573 km. It is noted that the slope of linear fitting line is smaller than 1:1 line, indicating that $MLH_{RS}$ tends to be larger than $MLH_L$ in the afternoon, which is consistent with the case study. Although the comparison only exist in summer, it can be generally concluded that $MLH_L$ from lidar in the afternoon characters the CBL height with good accuracy. As shown in the Fig.S4, the correlation of $MLH_L$' and $MLH_{RS}$ at 1400 LST is 0.330 and the value of MLH' is generally lower than $MLH_{RS}$, indicating overall improper for MLH' to describe the CBL height in the afternoon.

In fact, it would not exist complete agreement between $MLH_L$ ($MLH_L$') derived from lidar and $MLH_{RS}$ from radiosonde associated with several reasons. First, the two systems measure different atmospheric parameters (aerosol for lidar and temperature, humidity, wind for radiosonde) with varying height resolution and accuracy and these parameters are influenced in different way by the processes occurring within PBL (Seibert et al., 2000). Additionally, it is difficult to identify a clear upper boundary of the mixing layer because the measured parameter is actually not a fixed point but rather a transition layer between two atmospheric states (Stull, 1988; Garratt, 1992; Collaud et al., 2014).

A data set containing nearly six years measurement in Beijing is used for assessment of the overall performance of the Wavelet MLH algorithm ($MLH_L$ and $MLH_L$') with respect to the diurnal availability, as shown in Fig. 7. It can be seen that the expected shape presenting the growth of a convective mixing layer is observed. Owing to solar heating of the surface, when convective layer begin to rise due to upward convection in the early morning and nocturnal residual layer tends to collapse, $MLH_L$ from lidar presents the minimal value. After that, $MLH_L$ grows continuously and reaches its maximum height around 1500, with the value of 1.449, similar to results found for Vienna (Lotteraner and Piringer, 2016) and Berlin (Geiß et al., 2017). The shaded areas indicate the temporal variability as calculated from the standard deviation of $MLH_L$ ($MLH_L$'). It is on the order of 600 m for $MLH_L$ (300 m for $MLH_L$') during 0900 -1500 LST, while the other period is on the order of 700 km (400 m for $MLH_L$'). The larger standard deviation is attributed to the variability of residual layer.

The diurnal cycles derived from $MLH_L$ match well with RS results in the afternoon, but larger than $MLH_{RS}$ in the early morning and evening. Contrarily, $MLH_L$' tend to approach to $MLH_{RS}$ in the early morning and evening, but keep far from $MLH_{RS}$ in the afternoon. The difference between $MLH_L$ (1.405 ±0.675 km, mean ± standard deviation of the mean) and $MLH_{RS}$ (1.524 ±0.582 km) at 1400 is around 0.120 km. It is reasonable considering that RS data is acquired at 1400 only from summer time, when $MLH_L$ is usually larger around the year. However, the discrepancy between $MLH_L$' (0.912 ± 0.315 km) and $MLH_{RS}$ (1.524 ±0.582 km) at 1400 is around 0.612 km. Actually, $MLH_L$ is nearly 0.46 km larger than $MLH_L$' throughout the day, and with bigger standard deviation. As to the measurement at 0800 LST, the difference between $MLH_L$ (1.196 ±0.710 km) and $MLH_{RS}$ (0.434 ±0.364 km) is around 0.762 km, which is larger than the difference between $MLH_L$' (0.755 ± 0.334 km) and $MLH_{RS}$ (0.434 ±0.364 km). The discrepancy of lidar and RS measurements at 2000 LST is similar.

Overall, $MLH_L$' can catches the high SBL height in the nocturnal time, when it is larger than 300 m. The stable layer height detected by MLHL' in the night time is the layer in which ground-emitted atmospheric pollutants are trapped, it contributes to the assessment of the surface pollutant concentration when there is emission in the nocturnal time using the numerical models. (Collaud et al., 2014).Due to incomplete optical overlap, in some case the point derived from $MLH_L$' is residual layer height rather than the low nocturnal SBL height. And in the daytime, $MLH_L$' tends to be lower than CBL height. In the study of Mues et al. (2017) and Kotthaus et al. (2018), the MLH in the daytime is usually assigned as the lowest layer detected by ceilometer. Using the higher-power lasers (CE-370)with increasing SNR and small gradient detected, attribution of the lowest layer in the daytime may remain open, since the first local maximum gradient ($MLH_L$') not always corresponds to the biggest local maximum ($MLH_L$). Our study indicates that $MLH_L$ retrieves the consistent RL height during the night following the CBL diurnal maximal. The RL height corresponds to trapped atmospheric constituents discharged some hours before, which can be employed to convert column-mean optical depths into near-surface air quality information from remote sensing. And, the SBL height provided by radiosonde at 0800 and 2000 LST can be considered as complementary to the lidar approaches.

### 3.3. Climatology of MLH in Beijing

### 3.3.1. Seasonal variation

The seasonal mean diurnal cycle of the $MLH_L$ from lidar is shown in Fig. 8. An evident seasonal variation of magnitude of the diurnal cycle is observed. As shown in Table 1 and Fig. 8, the smallest $MLH_L$ magnitude is found in winter with the peak value of 1.404 ±0.751 km at 1500 LST, whereas spring demonstrates the maximum magnitude with 1.647 ±0.754 km at 1500 LST. The maximum in summer is 1.526 ±0.581 km, and the maximum in autumn is 1.445 ±0.837 km. From the all-day average of the four seasons, the averages in spring, summer, autumn and winter are 1.409 km, 1.261 km, 1.297 km and 1.228 km, respectively. In summer, $MLH_L$ acquired by lidar at 1400 LST and 1500 LST is 1.430 km and 1.507 km, respectively, while $MLH_{RS}$ at 1400 LST is 1.524 km. The measurement of $MLH_L$ at 1500 LST is closer to $MLH_{RS}$ at 1400 LS. This is consistent to the case study that it takes some time for aerosol to diffuse upward with the drive of thermal turbulence. As the statistic

variation, the values of autumn $MLH_L$ vary most nearly at each hour with the bigger standard deviation, indicating the great fluctuations in the long measurement period, while variation of summer $MLH_L$ values for most hours is relative stable.

It should be noted that summer exists the biggest amplitude of diurnal variation of $MLH_L$, with the deepest valley (0.93 km) increasing to the peak value of 1.51 km. Tang et al. (2016) indicate that the lower $MLH_L$ value for summer nights and early mornings is contributed to effect of the mountain plain wind. When the local mountain breeze from the northeast in the summer night superimposes the surface cooling, leading to the increase the thickness of the inversion layer, the height of the mixed layer gradually decreases. After sunrise, with the drive of thermal turbulence, the residual layer height observed by lidar is gradually replaced by a convective boundary layer height, with $MLH_L$ increasing rapidly, and after 12:00 LT, the plain wind from the south-westerly direction gradually dominates. Previous studies have suggested that the seasonal variation in the $MLH_L$ may be associated with radiation flux (Stull, 1988;Kamp and McKendry, 2010; Munoz and Undurraga, 2010), which was consistent with our results. The observational data from Tang et al. (2016) indicated that radiation flux of spring is more than that in summer. The relatively low values in autumn and winter are likely to relate to the low radiation flux.

### 3.3.2. Interannual variation

Interannual variations of $MLH_L$ diurnal cycle are investigated in Beijing from 2013 to 2018, as shown in Fig. 9. Diurnal variations of $MLH_L$ in different years all have same patterns, but with the different magnitude. Clearly, from 2013-2018, the values of diurnal circle $MLH_L$ increase year by year, including both the RL height at night and CBL height at daytime. Since the data of 2013 is mainly from winter and spring, the $MLH_L$ seems stable, not like the amplitude of other years. As shown in the Fig. 9 and Table 2, from 2014 to 2018, the $MLH_L$ all-day maximum values around 1500 LST grow year by year, with the value of 1.291 ±0.646 km, 1.435 ±0.755 km, 1.577 ±0.739 km, 1.597 ±0.701 km and 1.629 ±0.751 km, respectively. The all-day average of $MLH_L$ are 1.110 km, 1.216 km, 1.352 km, 1.391 km and 1.502 km, respectively. also showing an increasing trend. It indicates the volume available for the dispersion of pollutants extending, which is beneficial to the mitigation of surface pollution. As shown in Figure S5, from 2014 to 2018, the cumulative increase of the mean $MLH_L$ of the whole day was 0.392 km, the total increase of the maximum is 0.338 km. As for annual increase of $MLH_L$, the average of all-day increments in 2016 is the largest (0.136 km), while the average of all-day increments in 2017 is the smallest (0.039 km).

In particular, the interannual variation of $MLH_L$ in the period from 10:00 LST to 15:00 LST is calculated, when PBL is characterized by obvious convective boundary layer. From 2014 to 2018, the average CBL height show a significant increasing trend, which are 1.075 km, 1.212 km, 1.324 km, 1.351 km and 1.533 km, respectively. The total increase in the average CBL height is 0.458 km. As for annual increase in CBL height, the average increase in CBL height in 2018 is the largest (0.182 km), while the average increment of the whole day in 2017 is the smallest (0.027 km).

It is found that, based on the measurement of 2014 to 2017, $MLH_L$ has a strong negative correlation with AOD (R = -0.41) and with relative humidity (R = -0.21), while $MLH_L$ presents a positive correlation with wind speed (R = 0.43) and shows no

correlation with temperature (Fig. 10). As the study of Wang et al. (2019), from 2014 to 2017 in Beijing, AOD and surface PM$_{2.5}$ has a tendency to decrease year by year, while MLH$_L$ increases gradually. The reduction of AOD and surface PM$_{2.5}$ is revealed to relate to pollution emission control in recent years (Zhang et al., 2019). Compared with 2016, relative humidity has increased in 2017 and wind speed has weakened, showing no good to the development of MLH, which is consistent with the small increase in MLH$_L$ (0.027 km) in 2017. In addition to the effects of meteorological conditions, the increase of the MLH$_L$ benefits from the improvement of air quality in Beijing in recent years (Wang et al., 2019). Due to the scattering and absorbing of aerosol, the solar radiation received from ground decreases. It is thermal buoyancy generated from surface radiation that drive the PBL to develop. Thus, the development of MLH is suppressed under high aerosol load. Hence, with the relief of radiation effect by aerosol during these years, the turbulence increases, thus leading to larger PBL height (Ding et al., 2016; Li et al., 2017; Wang et al., 2019). MLH affects the concentration of pollutant near the surface, while total radiation of aerosol within the column atmosphere in turn influences the MLH.

### 3.4. Implication for surface pollution retrieval

The vertical structure of the ML is important for the pollution concentrations at the surface due to its impact on the volume into which pollutants are mixed. Mues et al. (2017) reported that black carbon concentrations show a clear anti-correlation with MLH measurements. Hu et al. (2014) found a negative correlation between near-surface O$_3$ and MLH for seven cities in the North China Plain. In the study, as shown in Fig. 11, the correlation between MLH$_L$ and observed PM$_{2.5}$ data from the same observatory shows high negative correlation (R=-0.569) with the four years measurement (2014-2017). Actually, the pollutant concentration near surface is affected by the overall effect of the local emission and meteorological condition, with variation of different spatio-temporal distribution. MLH is just one of these influencing factor. Geiß et al., (2017) indicated that when MLH and near-surface concentrations are linked, it is necessary to take the locations, i.e., meteorological conditions and local sources, and the details of the MLH retrieval into account. In fact, all the data used in our study is observed from the same observatory. And, PMRS model is used to calculate the surface PM$_{2.5}$ concentration includes the parameters of the emission (AOD) and meteorological condition (RH) into account.

Due to the difference in the source of the MLH, lidar and radiosonde, the comparison of derived PM$_{2.5}$_lidar and PM$_{2.5}$_RS with the in-situ observational PM$_{2.5}$ data at 0800 LST is presented in Fig. 12a and 12b. MLH from lidar shows reasonably good performance for the retrieval of PM$_{2.5}$ in the morning, with the correlation coefficients of 0.741 and RMSE 46.69 μg/m$^3$. However, the calculated PM$_{2.5}$ from MLH$_{RS}$ obviously overestimate the surface pollution, with lower correlation coefficients and larger standard deviation. The large overestimation should be contributed to the underrating of aerosol layer height. In the morning when the PBL is not well developed, above MLH$_{RS}$ there still exist a large amount of aerosol, referring the lidar images of Fig. 1- Fig. 4. The discrepancy makes sense using the method with the observed total amount of pollutant of the column atmosphere, including the emission from surface and the residual aerosol from the day before. Therefore, MLH$_L$ from

lidar, as the good indicator of the aerosol layer height, is more suitable for estimating the surface air pollution from the column-mean optical depths.

As presented in Fig.12c, the calculated PM$_{2.5}$_lidar data of daytime period (0800-1700 LST) shows higher correlation (0.846) than that of only the early morning, and the little larger RMSE (55.58 μg/m$^3$) is associated to the larger amount of samples into statistic. Considering the uncertainty of the series of parameters used in the model, the agreement between calculated PM$_{2.5}$_lidar and in-situ measurement is reasonable good. Actually, the accuracy also shows a diurnal cycle, with the peak of correlation coefficients (0.927) at 14 LST (Fig.12d). The correlations at 12, 13, 14, and 15 LST were 0.894, 0.922, 0.927, and 0.900, respectively. The higher accuracy may be due to the completed mixing of the aerosol at noon and the vertical distribution of the aerosol tend to be uniform. The correlation between 8, 9 and 17 LST is less than 0.8, which is related to the complex boundary layer structure in the morning and nightfall. It is difficult to achieve fully mixing of the aerosol in the stable boundary layer or the residual layer. The smaller RMSE is related to the limited samples. Therefore, the daily variation of calculated surface pollutant accuracy using MLH retrieval by lidar vary with the daily variation of aerosol mixing uniformity at different times during the daytime. Based on the observational data, PM$_{2.5}$ tends to peak in the morning and evening. In contrast, afternoon usually witnessed lower mass concentration due to rapid vertical diffusion of aerosols (Guo et al., 2016).Thus, MLH$_L$ from lidar can offer the significant contribution to retrieve the diurnal circle of the surface air pollution.

## 4. Summary and conclusions

To acquire the high-resolution observations of MLH diurnal variation, a study using lidar was performed from January 2013 to December 2018 in the Beijing urban area. Detection of the MLH based on two wavelet methods (MLH$_L$ and MLH$_L$') applied to lidar observations is operated. The two data results are compared with radiosonde as case studies and statistical forms. The temporal resolution (two or three measurements per day) of PBL detection by RS is not able to provide the mixing layer height diurnal cycle, no matter its good precision. MLH shows good performance for the convective layer height at daytime and the residual layer height at night. MLH$_L$' have the potential to describe the stable layer height at night sometime, even though the capability is limited due to the high incomplete overlap of lidar used in the study. The stable layer height detected by MLH$_L$' in the nighttime is the layer in which ground-emitted atmospheric pollutants are trapped, it contributes to the assessment of the surface pollutant concentration when there is emission in the nocturnal time using the numerical models. Whilst the residual layer height corresponding to trapped atmospheric constituents discharged some hours before, which can be employed to convert column-mean optical depths into near-surface air quality information from remote sensing. And, MLH$_L$' does not always work out to catch the convective layer height as MLH$_L$ in the afternoon. Nevertheless, MLH$_L$' could be a useful complementary as stable layer height for dataset of MLH$_L$ in some cases.

Nearly six year climatology for $MLH_L$ diurnal cycle is calculated for convective and stable conditions. It is true that the height of mixing layer obtained by different approaches may be different. We focus on the temporal change of aerosol layer height with a consistent method using the dataset of $MLH_L$. The maximum $MLH_L$ characteristics of seasonal change in Beijing indicate that it is low in winter (1.404 $\pm$ 0.751km) and autumn (1.445 $\pm$ 0.837km), and high in spring (1.647 $\pm$ 0.754km) and summer (1.526 $\pm$ 0.581km). A significant phenomenon is found that from 2014 to 2018, the magnitude of diurnal cycle of $MLH_L$ increase year by year. The cumulative increase of the mean $MLH_L$ of the whole day is 0.392 km, and the total increase of the maximum is 0.338 km. It may partly benefit from the improvement of air quality. As to converting the column optical depth to the surface pollution, the calculated $PM_{2.5}$ using $MLH_L$ data from lidar shows better accuracy than that from radiosonde, compared with observational $PM_{2.5}$. Additionally, the accuracy of calculated $PM_{2.5}$ using $MLH_L$ shows a diurnal cycle in the daytime, with the peak at time of 14 LST. For the operational measurement of PBL height, MLH from lidar has the capability to mark the diurnal circle of mixing layer height, and can be used as an effective parameter for the vertical distribution of aerosols, providing an important reference to obtain near-ground pollutant concentrations for remote sensing. Actually, interpreting data from aerosol lidar is often not straightforward, because the detected aerosol layers are not always the result of ongoing vertical mixing, but may originate from advective transport or past accumulation processes (Russell et al., 1974; Coulter, 1979; Baxter, 1991; Batchvarova et al., 1999; Tang et al., 2016). Each detection method has good performances only for defined ML structures and under specific meteorological conditions. Therefore, combination of several methods and instruments may contribute to characterize the complete diurnal cycle of the complex ML structure (Wiegner et al., 2006; de Bruine et al., 2017; Morille et al., 2017; Kotthaus et al., 2018 ).

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

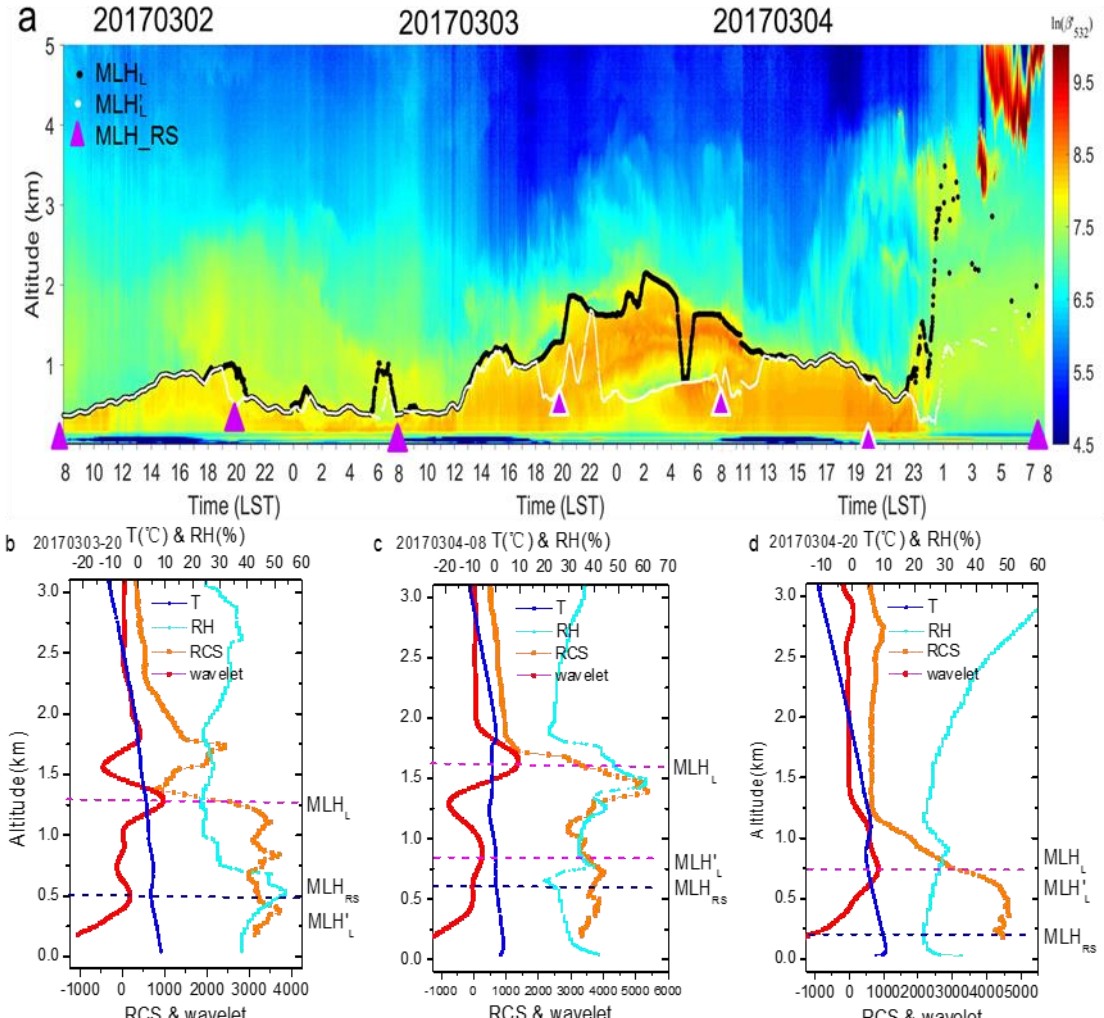

Fig. 1. Upper: (a) Daily backscatter profiles from lidar for 20170302-20170304 cases. The lines connected by black dots in the figure represent the retrieved MLH$_L$, while white dots line indicate the MLH$_L$' and the top of purple triangle indicate the MLH$_{RS}$ identified from radiosonde. The horizontal axis represents the local standard time(LST) and the vertical axis represents the height. Colorbar denotes the logarithm of the attenuated backscattering coefficient (In($\beta'_{532}$)).

Lower: The vertical profile of RCS (orange curve) from lidar and wavelet coefficient (red curve) of RCS, as well as the vertical profile of temperature (T) (blue curve) and relative humidity (RH) (cyan curve) for time of (b)20170303-20、(c)20170304-08 and (d)20170304-20 indicated by the white edge triangle in the upper picture (a).

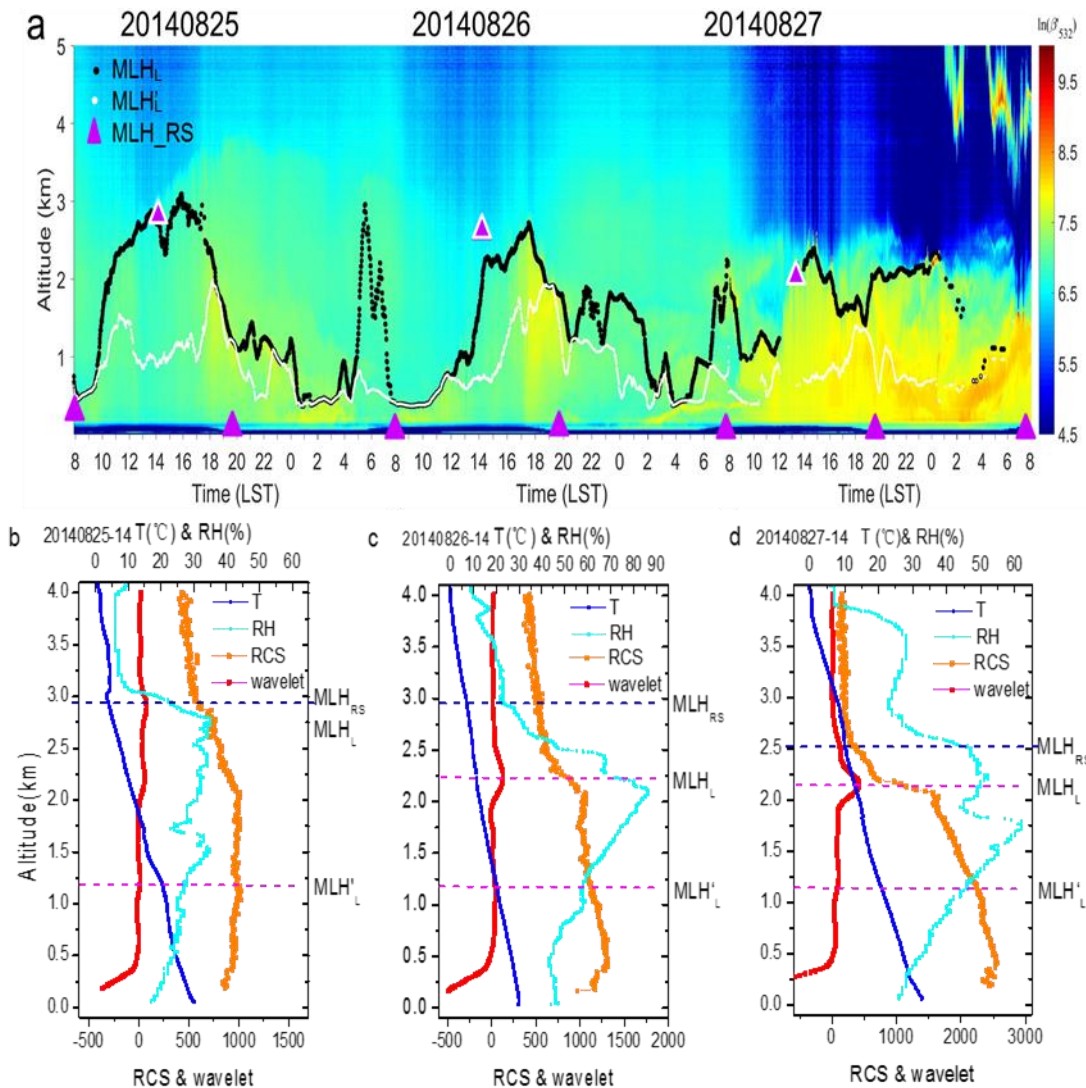

Fig. 2. Similar as Fig.1, but for (a)20140825-20140827 case, and vertical profile for (b)20140825-14、(c)20140827-14 and

(d)20140827-20.

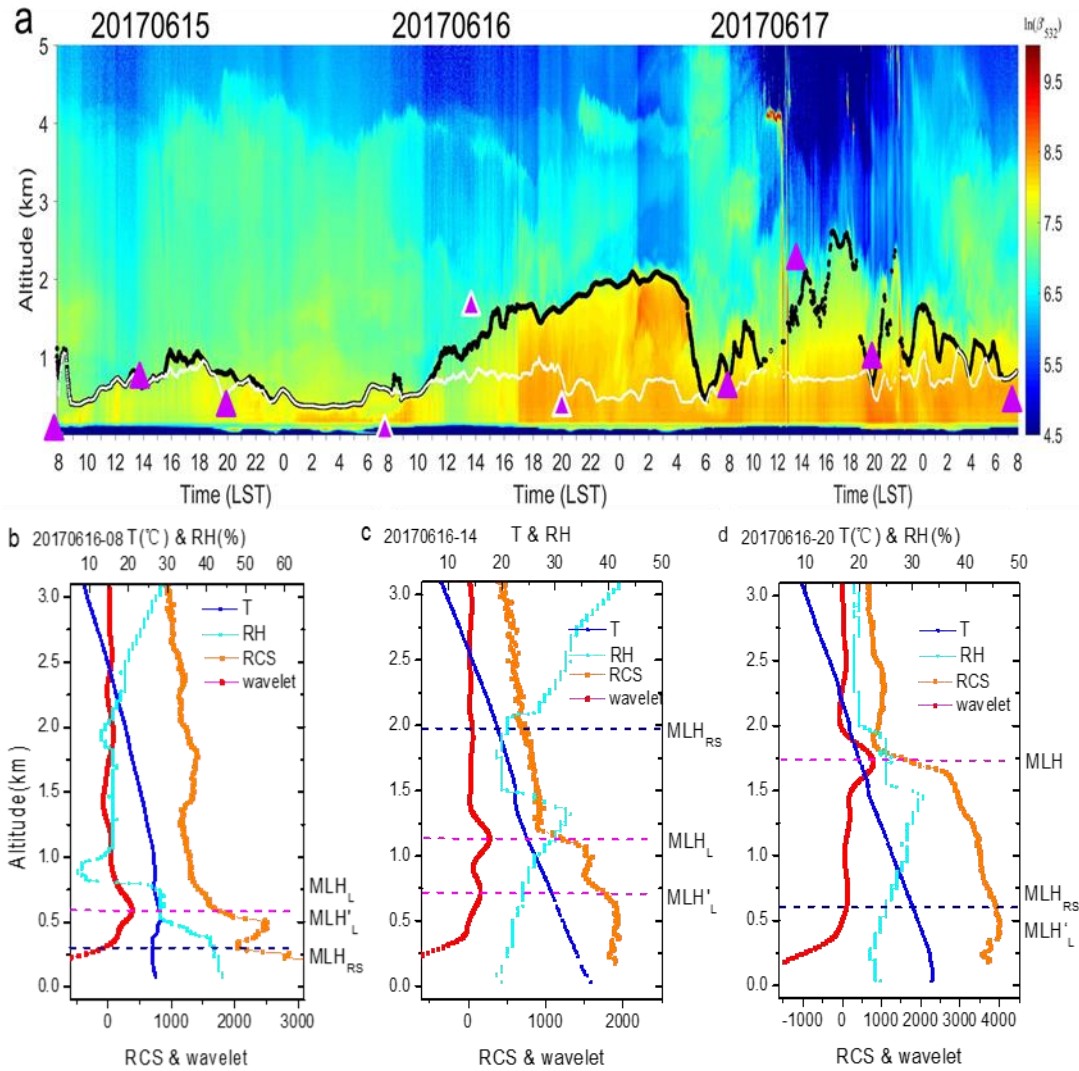

Fig. 3. Similar as Fig.1, but for (a)20170615-20170617 case, and vertical profile for (b)20170616-08、(c)20170616-14 and (d)20170616-20.

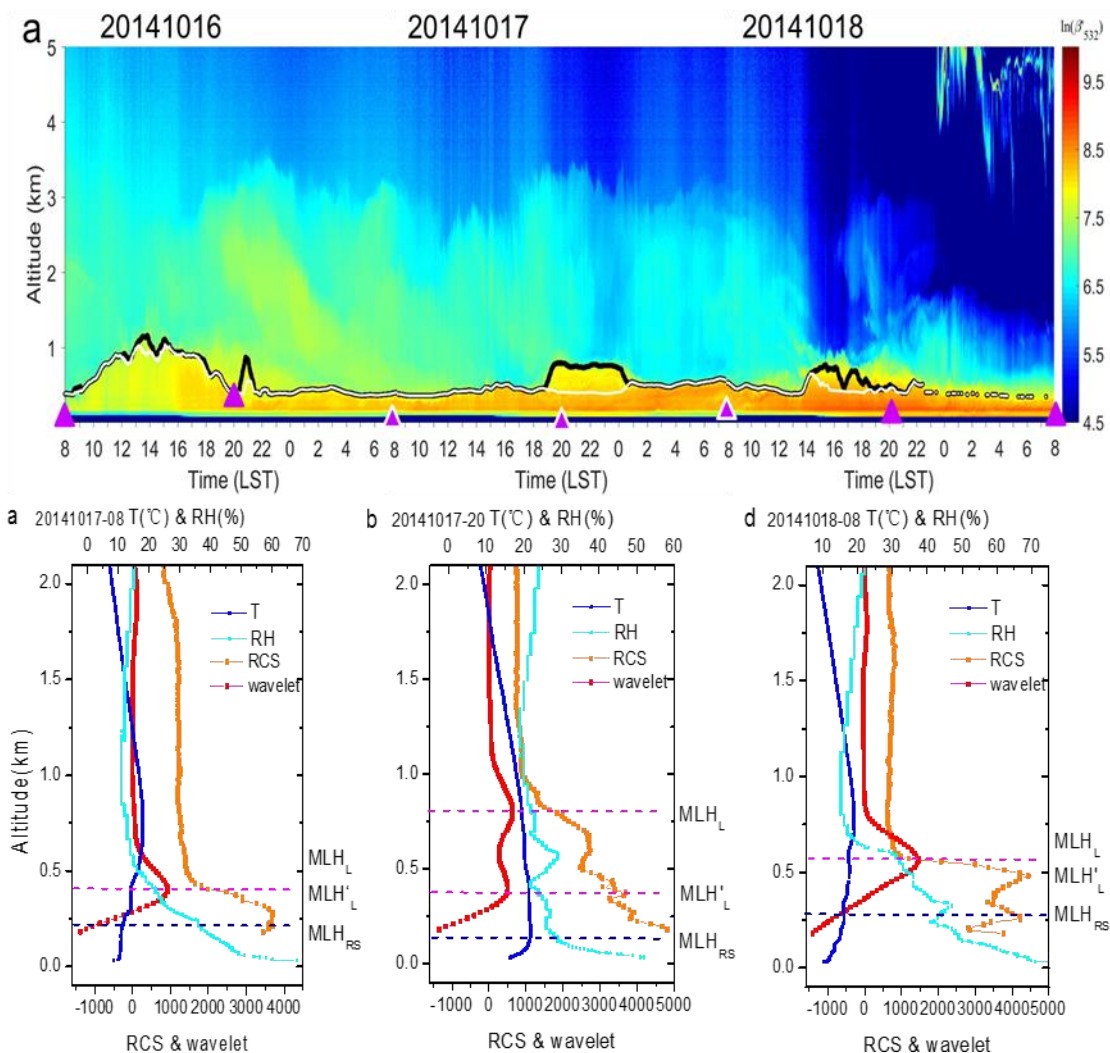

Fig. 4. Similar as Fig.1, but for (a)20141016-20141018 case, and vertical profile for (b)20141017-08、(c)20141017-20 and (d)20141018-08.

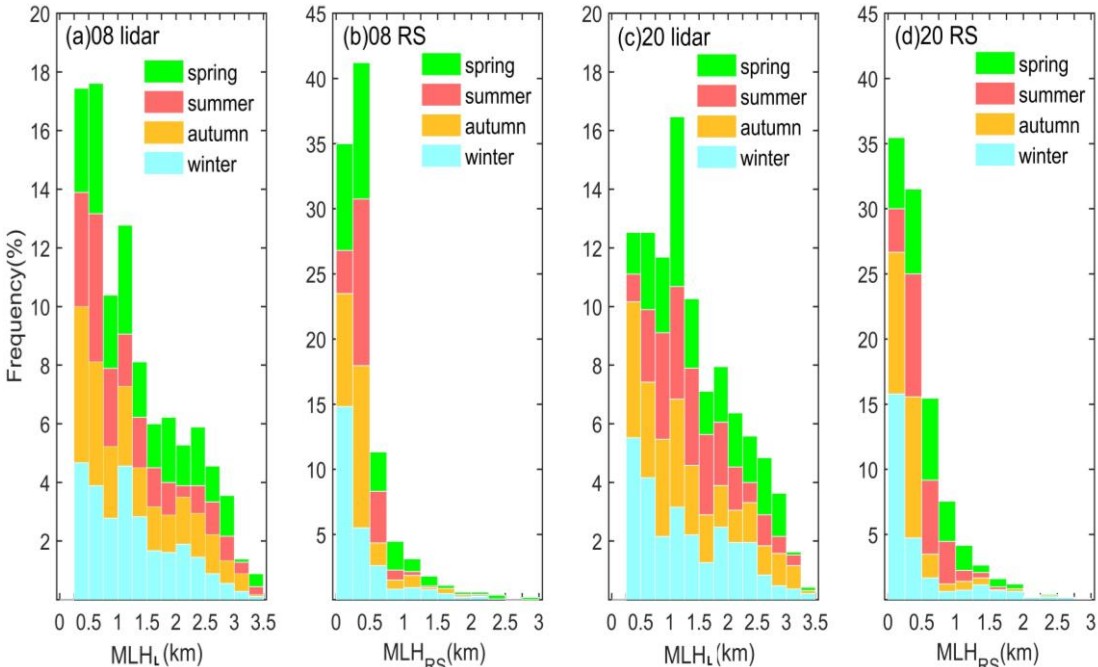

Fig. 5. Comparison of frequency distribution of all MLH$_L$ (2013-2018) retrieved from lidar and MLH$_{RS}$ from radiosonde with the supplementary information of seasonal variation. MLH from (a) lidar and (b) radiosonde at time of 08 (LST), (c) lidar and (d) radiosonde at time of 20 (LST) are presented. Noted that for presenting the detail distribution, MLH$_L$ adds up to 20%, while MLH$_{RS}$ add up to 45%.

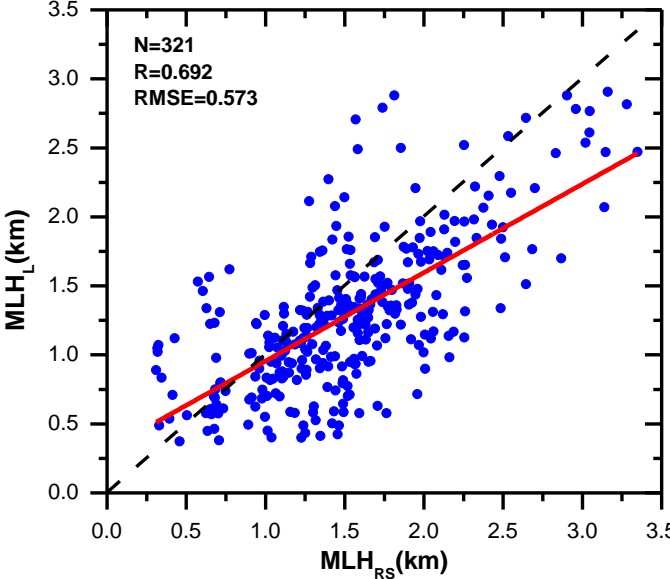

Fig. 6. Comparisons between $MLH_L$ in summer derived from lidar and $MLH_{RS}$ from radiosonde at time of 14 (LST). Red line

indicates the linear fitting of 321 samples, while the black dash line represents the 1:1 line.

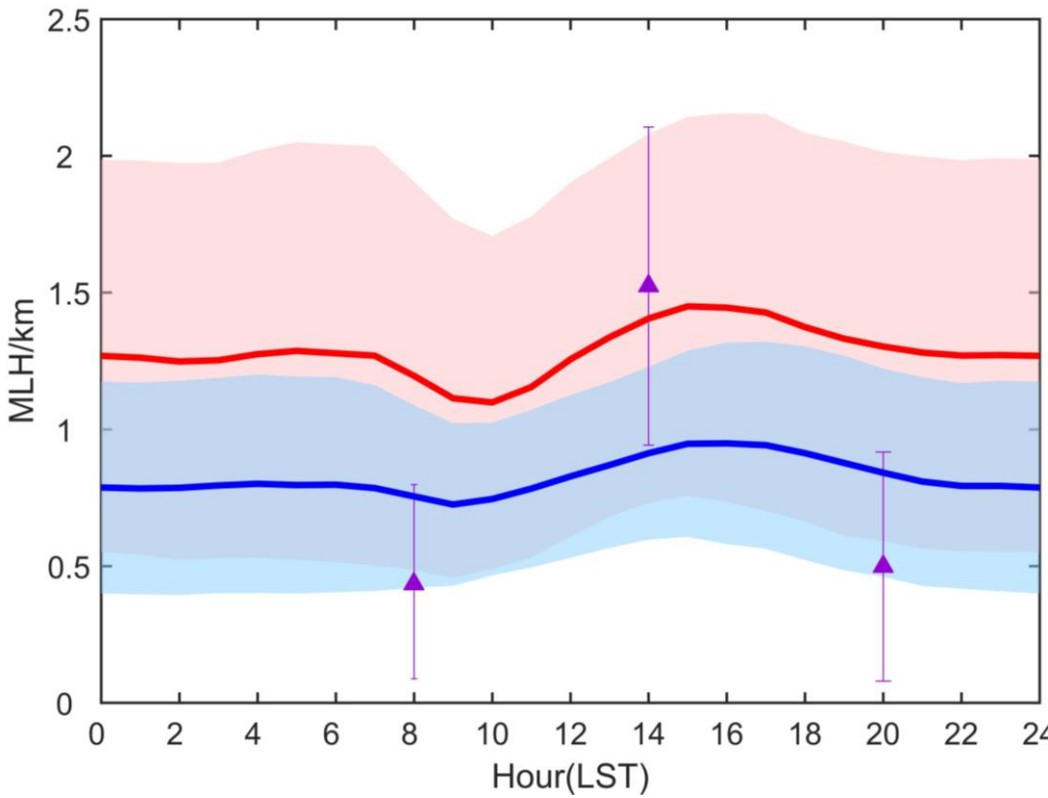

Fig. 7. Diurnal cycles of mixing layer height. The red line indicates the $MLH_L$ retrieved from lidar, and the blue line

represent the $MLH_L$' from lidar. The shaded areas show the standard deviation of $MLH_L$ and $MLH_L$'. Purple triangles

indicate the $MLH_{RS}$ averaged from routine RS data at 08 and 20 time (LST), and from summer radiosonde at time of 14

(LST). The purple lined indicate the standard deviation of $MLH_{RS}$.

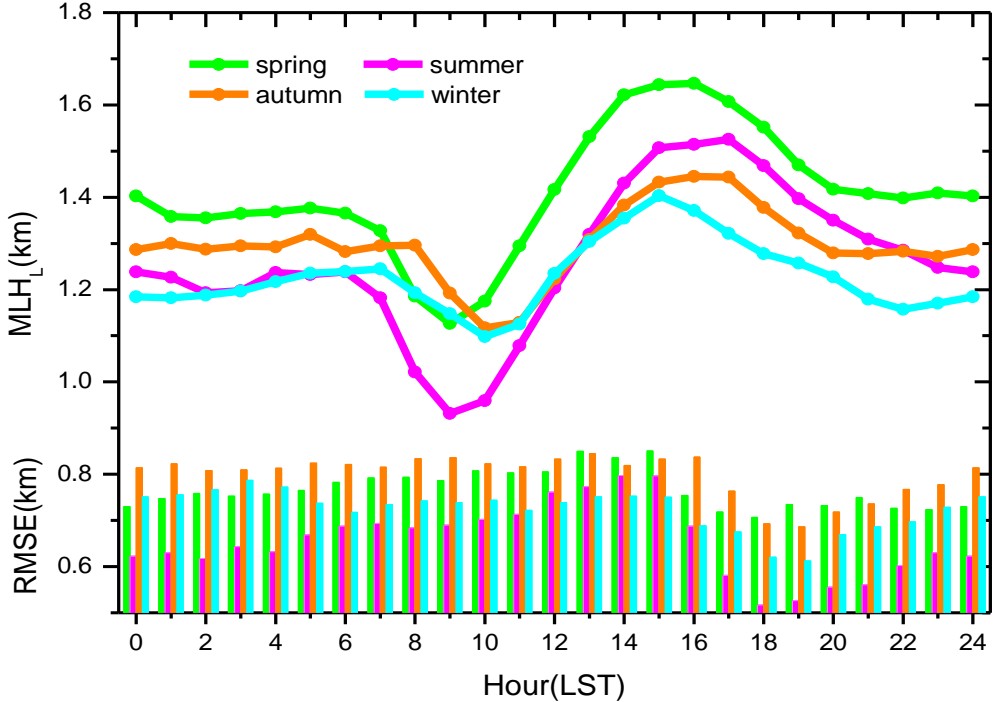

Fig. 8. Seasonal variation of diurnal cycles of $MLH_L$ retrieved from lidar (dot lines), as well as the standard deviation of $MLH_L$ (histograms). The red triangle indicates the $MLH_{RS}$ measured at 1400 LST by radiosonde.

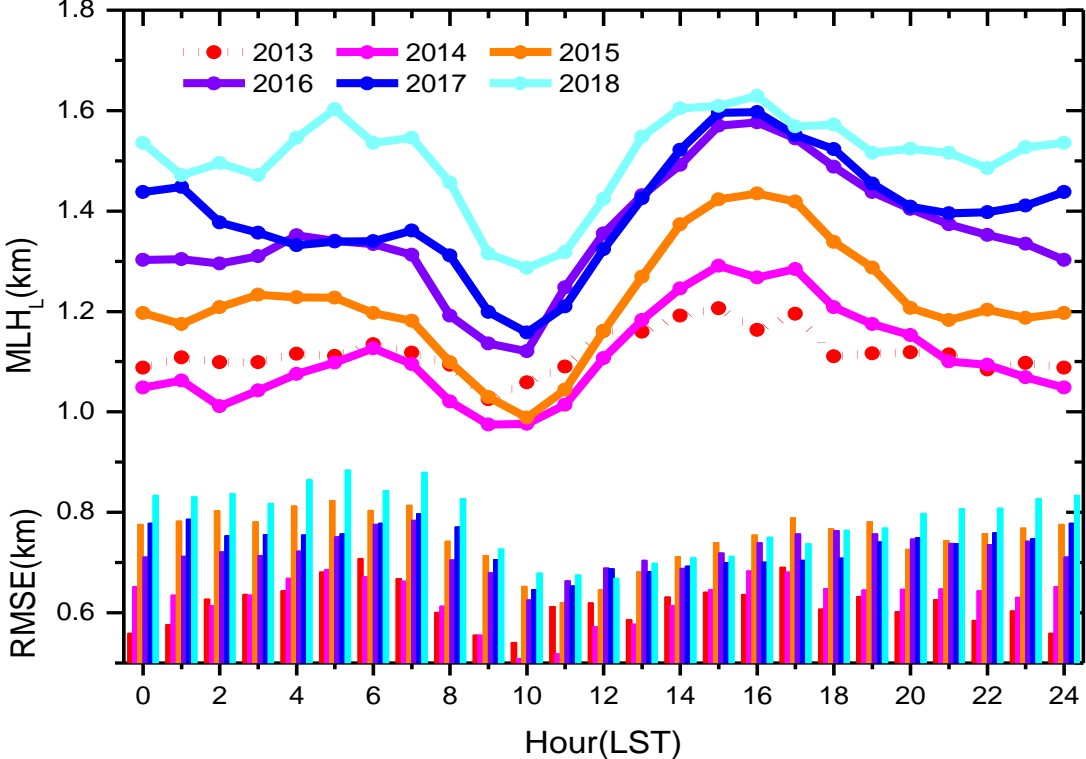

Fig. 9. Interannual variation of diurnal cycles of averaged $MLH_L$ retrieved from lidar (dot lines), as well as the standard deviation of $MLH_L$ (histograms). Due to the incomplete data of 2013, the $MLH_L$ data of 2013 is presented as dot line to be noted.

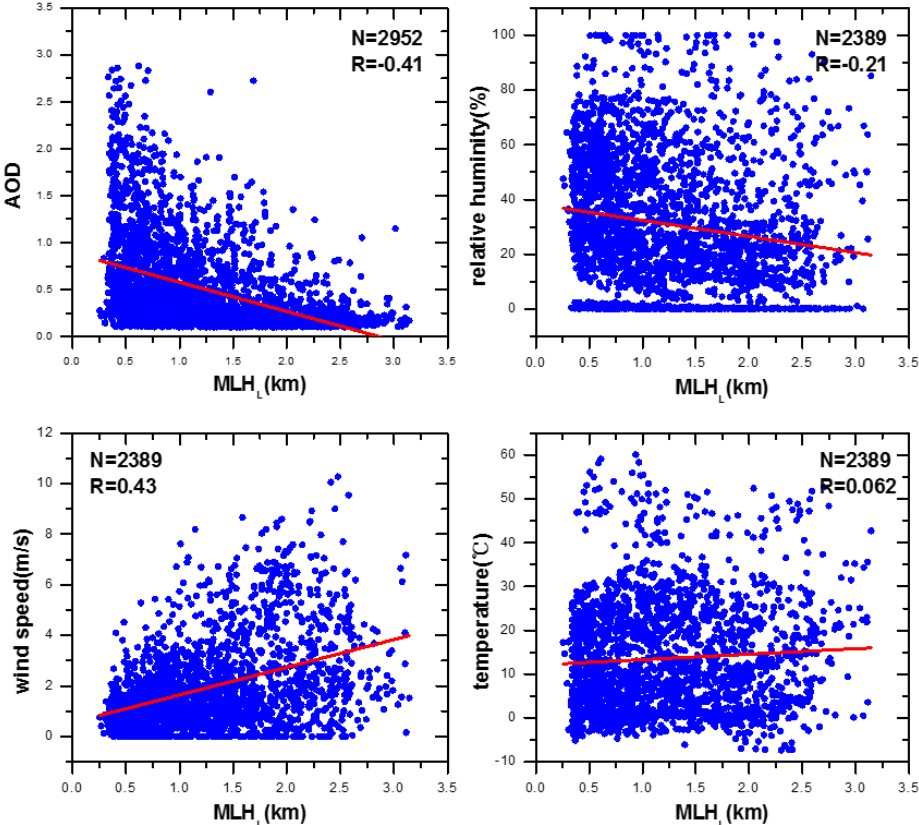

Fig.10. The correlation between MLH$_L$ and AOD, relative humidity, wind speed and temperature with the measurement from
2014 to 2017.

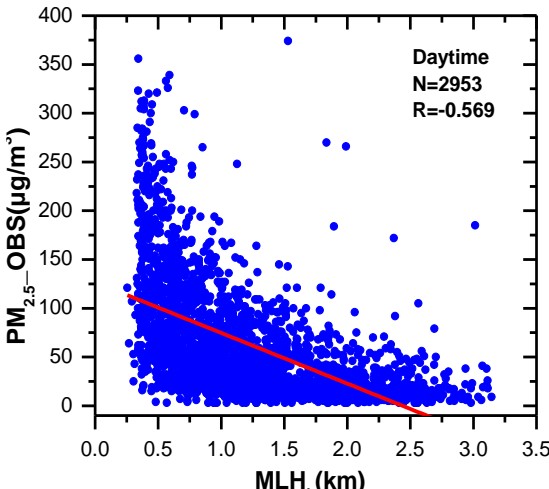

Fig.11. The correlation between MLH$_L$ and PM$_{2.5}$_OBS with the measurement from 2014 to 2017.

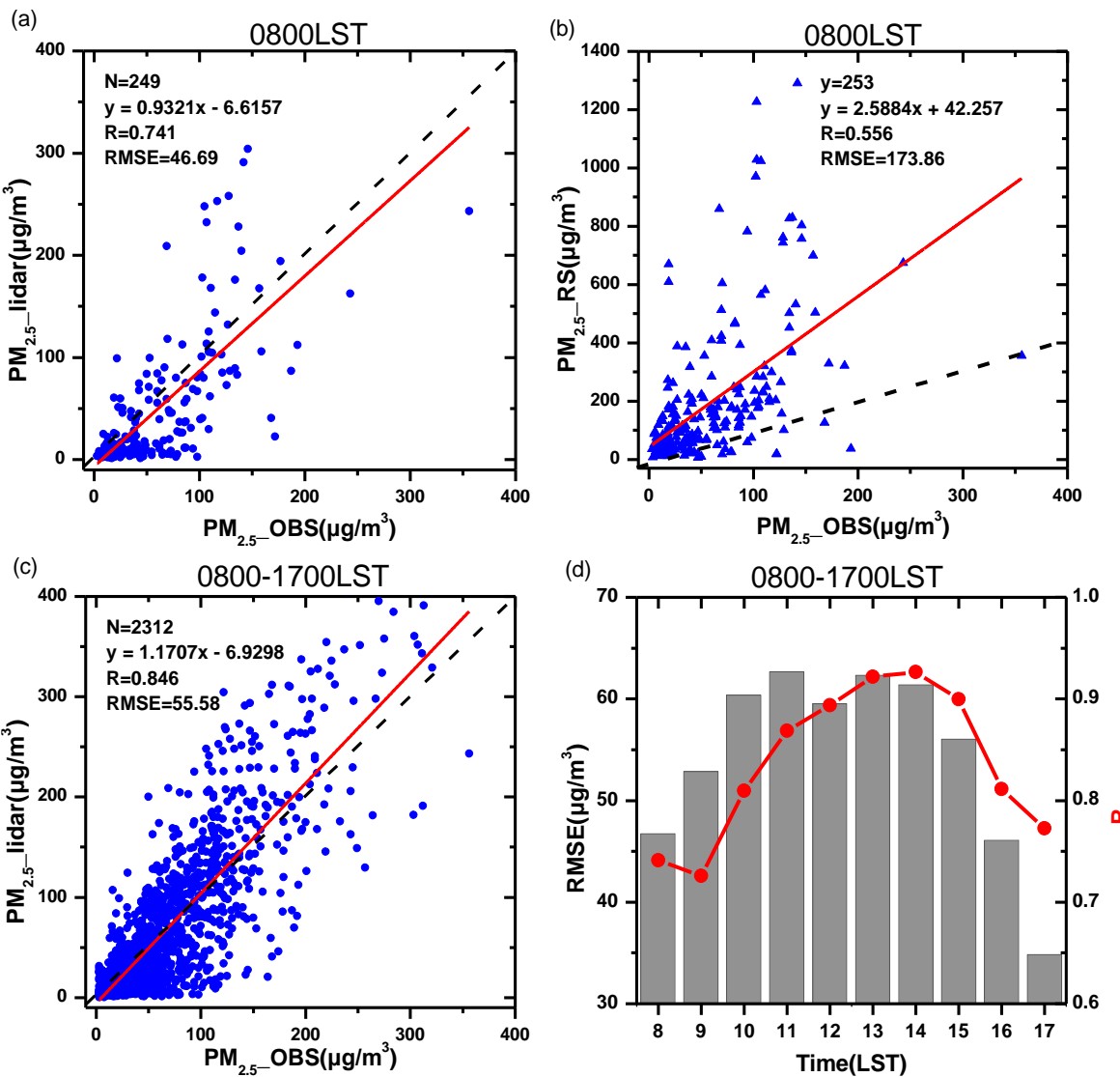

675

Fig. 12. Comparisons between observed $PM_{2.5}$ and $PM_{2.5}$ calculated from PMRS model using (a) $MLH_L$ and (b) $MLH_{RS}$ at time of 0800(LST), and (c) $MLH_L$ for the period of 0800-1700(LST) , and (d) correlation coefficient and RMSE between observed $PM_{2.5}$ and $PM_{2.5}$ calculated from PMRS model using $MLH_L$ for each hour from 0800 to 1700 LST.

Table 1 Statistics of boundary layer height seasonal change

| $MLH_L$/km | Spring | summer | autumn | winter |
|---|---|---|---|---|
| mean | 1.409 | 1.261 | 1.297 | 1.228 |
| maximum | 1.647 | 1.526 | 1.445 | 1.404 |
| minimum | 1.126 | 0.932 | 1.117 | 1.098 |

Table 2 Statistics of boundary layer height interannual change

| $MLH_L$/km | 2013 | 2014 | 2015 | 2016 | 2017 | 2018 |
|---|---|---|---|---|---|---|
| mean | 1.118 | 1.110 | 1.216 | 1.352 | 1.391 | 1.502 |
| maximum | 1.207 | 1.291 | 1.435 | 1.577 | 1.597 | 1.629 |
| minimum | 1.025 | 0.975 | 0.989 | 1.121 | 1.158 | 1.287 |