# Peer review of "Determination and climatology of diurnal cycle of atmospheric mixing layer height over Beijing 2013-2018: Lidar measurements and implication for air pollution"

_Atmospheric Chemistry and Physics, 2020_

## Short Comment (SC2)

**Reply to Anonymous Referee #2:**

We appreciate the reviewer's comments on the manuscript. We have studied them carefully. Below are point-by point responses to the referee's comments.

**Comments from the editors and reviewers: General comments:**

This manuscript reports climatological values of mixing layer height (MLH) over Beijing for from Lidar measurements, their evaluation against radiosonde based MLH calculation and argues that MLH values derived from Lidar is better than that from RS for estimation of PM2.5 at surface. The manuscript needs a lot of improvement before publication. Please see comments and suggestions below:

 The first part of the results and discussion, where the comparison of the various approaches to estimate MLH is presented is not satisfactory. First, the English language is poorly written which makes it hard to understand what is being conveyed. Also, the text corresponding to Figures 2-4 lacks any discussion of the features seen in them. More detailed discussion and language improvement is needed.

**A1: This part has been rephrased, please see the revised manuscript.**

2. Fig 3d: why is MLH\_RS detected at 0.6 km? it should be 2km as seen in Fig 3c.

**A2:** The vertical profile present is actually indicated by "the white edge triangle in the upper picture", also the measurement time can be seen on the top of Fig. 3c (20170606-14) and Fig. 3d (20170606-20). Fig. 3d show MLHRS (0.6 km) at 2000 LST, while Fig.3c indicates (2 km) at 1400 LST. Due to the different time, MLHRS varies.

3. Actually, the whole context of the first 4 figures is not understood. Why are these days shown here? why not any other day? Is it meant to show seasonal variability i.e one for one season? If the figures are introduced to show various types of differences between the 3 methods, then the discussion show be organized in that manner and the inter comparison figure 5 should be discussed in context of features seen in these figures 1-4. Better organization is needed.

**A3:** Fig. 1-Fig. 4 present as case study is aimed to show the evolution of the MLH and comparisons of lidar measurement (MLHL and MLHL') and RS measurement (MLHRS). The four cases is almost all cloud free to present continuous retrievals. Fig.1 shows obvious separated layers in the evening (spring without 1400LST measurement). Fig.2 shows low aerosol load, in which condition lidar can still capture the small gradient (summer with 1400 LST data). Fig. 3 relative high aerosol load and MLHL and MLHL' show obvious difference in the afternoon (summer with 1400 LST data). Fig. 4 keep stable though the whole day. The information delivered of the figure are discussed in the following context in the revised manuscript.

4. Please give more details about the cloud screen/flag used (Line 111). How was cloud detected, at what resolution etc?

**A4:** The content is added that "a threshold is selected to distinguish between clouds and aerosol layers." and the lidar "can detect a long range profile up to 30 km every 1 second. For the enhancement of signal noise ratio, 60 profiles are averaged to restore as one with the time resolution of 1 minute." "For convenient comparison with air quality and meteorological parameters, all MLH results are one hour averaged."

5. Fig 1a, 2a, 3a, 4a all are stretched, the fonts are of different sizes compared to other panels and should be made consistent. Figure 1b have an undefined variable MLH2 in the figure label?

**A5:** The layout of the sub figure is based on requirement of clarity, expressiveness and artistry. In our view of Fig. 1a-4a, the information conveyed changes nothing. In Figure 1b, MLH2 has been revised to  $MLH_L'$ , which indicates the first local maximum of lidar MLH.

6. Evaluation of MLH\_lidar is essential for second half of this manuscript. Hence, more detailed comparison and discussion is necessary. scatter plots in supplementary should be presented in main and discussed in detail. Moreover, as seasonal variability is significant on MLH, the comparison in Figure 6 should include and discuss scatter plots for all the seasons, separately.

**A6:** The comparisons of MLH from lidar (MLHL and MLHL') and MLHRS is the one of objective of the study. From Fig. 5, we can see that about 35% SBL is not at the lidar detection range for 0800 and 2000 LST. It can be concluded that the agreement of MLH from lidar (MLHL and MLHL') and MLHRS is poor, even though the scatter plots is not shown. The comparisons between MLHL' and MLHRS in the afternoon can be seen from the case study and the diurnal cycle, which indicates MLHL' is much lower than MLHRS. Please see the revised manuscript. Actually, the data of radiosonde is only available in summer, so the comparisons of the other seasons cannot be conducted.

7. From Fig 1-5, MLH from Lidar is termed as MLH but in Fig 6, it is termed as MLH\_Lidar. Please be consistent.

**A7:** All the biggest local maximum MLH of lidar is represented by  $MLH_L$ , while the first local maximum is indicated by  $MLH_L'$  and radiosonde result by  $MLH_{RS}$ . They are consistent in through the revised manuscript.

8. Fig 9: Is the MLH from RS also showing yearly variability similar to the MLH derived from Lidar? Please include the same from RS also in this figure and discuss.

**A8:** From 2014 to 2018, MLHRS at 0800 LST is 0.402, 0.412, 0.453, 0.444, 0.451 km, respectively. MLHRS at 2000 LST is 0.445, 0.501, 0.512, 0.515, 0.480 km, respectively. Both of the two measurement time show increasing trend. Due the availability of RS measurement is only in summer, it cannot compare with the annual mean value of lidar. We can see the annual variation of RS is not exactly the same as lidar. Actually, in the revised manuscript, the compassion of lidar

and RS measurement is discussed it the prior section, and in the section, we focus to present the annual variation of diunal cycle of MLH from lidar, while the RS measurements is just time points result.

9. In Figure 10d shows that correlations are highest when RMSE is also highest, this is non intuitive, please describe this feature. Also, the comparison in Figure 10 should be presented separately for each season as MLH has strong seasonal variations.

**A9:** The comment is added in the revised manuscript that "The smaller RMSE is related to the limited samples." And "larger RMSE is associated to the larger amount of samples into statistic." In the revised manuscript, there is already a lot of content. The seasonal variations of calculated PM2.5 and in-situ can be shown in the next paper.

10. Throughout the manuscript standard deviation and RMSE is used interchangeable, which should be corrected. They are not same.

**A10: Thank you for reminding, and the standard deviation has been revised to RMSE.**

11. Please check carefully for the typos in the manuscript. The title itself has one "airpollution"

**A11:** Thank you for reminding and I have noticed the typos. Hope the staff member of ACP could help to correct it. Actually, in my manuscript, it is right "air polllution".

---

## Referee Comment (RC1) · Anonymous Referee #1 · 13 Apr 2020

This manuscript analyses long-term measurements of mixed layer height (MLH) over Beijing. Authors describe and evaluate the techniques, derive climatological diurnal variation, and presents an application towards the estimation of fine particulate matter. Several comments and suggestions are offered for authors to consider while revising the manuscript for ACP.

Introduction section should provide more background, based on studies comparing mixed layer measurements, not limited to LIDAR but also from RADAR and other instruments. Some discussion has been made on the importance of MLH in context of

air pollution mixing and dispersion, which should be corroborated with relevant recent references (e.g. Singh et al., 2016; Mues et al., 2017).

Stronger correlations between LIDAR and Radiosonde are seen during afternoon but such correlations are absent in morning and evening. Besides poor correlation, values of MLH also do not match with radiosonde in morning and evening. There should be more deeper analysis and discussions on these aspects with references to previous studies.

The correlation analysis between MLH' and radiosonde should be shown for all three times in the supplement.

Section 3.1 describes mostly the variations as retrieved with limited new insights into boundary layer evolution. Additionally, several general statements are made e.g. "MLH' sometime agree well with SBL". Remove general statements and provide more specific discussions based on analysis.

Figure 8 shows significant reduction in MLH after sunrise, particularly during summer (l.245). This should be elaborated. How do the horizontal winds change during this time of minimum MLH? Have any other studies reported such variability in Beijing or elsewhere?

Interannual variation – It seems that some of the years have data limited to particular season (s). it will be appropriate to compare the years which have consistency in the seasonal coverage. Otherwise better to analyse a particular season among different years. In any case it is not clear what new is learnt by this analyses. There should be supporting analyses of temperature /winds and /or aerosol changes to explain observed inter-annual variations.

There are several variables defined MLH, MLH', MLH_RS etc. Try to use these variables consistently. For example in discussions MLH_RS is used but in Fig S3 -axis title -it is written as MLH.

Correlations are very weak r = 0.01 between radiosonde and lidar at 8LST (Fig S4) and data has significant scatter. Add some inter-comparison studies from literature to elaborate on this.

Computation of PM2.5 should be part of the main manuscript, instead of supplementary material.

Fig 7: show variability in RS data too.

Manuscript needs careful proofreading for language as various places. e.g. l.69: change "consistent" to "consistency" l.107: "MHL" to "MLH" l.118: check the sentence: "..where was located", probably it should be "which was located" l.195: change "collapse" to "collapsed" and "develop" to "developed" l.203: "shown" to "shows"

References Mues, A., Rupakheti, M., Münkel, C., Lauer, A., Bozem, H., Hoor, P., Butler, T., and Lawrence, M. G.: Investigation of the mixing layer height derived from ceilometer measurements in the Kathmandu Valley and implications for local air quality, Atmos. Chem. Phys., 17, 8157–8176, https://doi.org/10.5194/acp-17-8157-2017, 2017.

Singh, N., Solanki, R., Ojha, N., Janssen, R. H. H., Pozzer, A., and Dhaka, S. K.: Boundary layer evolution over the central Himalayas from radio wind profiler and model simulations, Atmos. Chem. Phys., 16, 10559–10572, https://doi.org/10.5194/acp-16-10559-2016, 2016.
* * *

---

## Short Comment (SC1) · 12 May 2020

Matthias Wiegner

m.wiegner@lmu.de

For the time being there is one reviewer's comment available. I agree with his/her general comments and suggestions so it is not necessary to repeat their remarks. Below please find a few additional comments focusing on selected technical or scientific details. As I am not an assigned reviewer I have only focused on the air quality aspect and the meaning of MLH'. Anyway, I hope that they can contribute to improve the paper.

- Title: Mentioning "implications for air pollution" in the manuscript's title is a little bit

misleading. The authors describe a methodology to derive the mixing layer height (convective, stable, residual layer (CBL, SL, RL)) from lidar measurements. They demonstrate that the agreement with radiosonde based retrievals is often quite limited. MLH is used as input for a numerical model (i.e. one equation) to estimate ground-based PM2.5 concentrations. The agreement with in-situ measurements is also limited. Consequently the whole procedure results in a rough estimate of PM2.5 only. Having this uncertainty and the heterogeneity of local sources of pollutants in mind (see also comments below), I feel that mentioning it in the title of the paper is somewhat exaggerated.

- L28: "...identification of pollutant emissions and sources": This statement seems to be misleading. The MLH (among others) has an influence of the dispersion of pollutants, but how sources can be identified from the MLH is not obvious and must be explained if the authors insist on that statement.

- L33: "contributes to the assessment of the pollutant concentration near the surface": I agree that MLH "contributes" to concentrations of pollutants but it must be kept in mind that the spatio-temporal distribution of sources plays a dominating role; see Geiß et al., (2017).

- L38: "MLH can be estimated by ... the concentration of PBL constituents". It is rather the other way around (having in mind the inherent problems already mentioned).

- L59: Here ceilometers should be explicitly mentioned. Lidars are comparably rare instruments, and only a few can be operated unattendedly and fully automatic (MPL is one of them, so often called a ceilometer or an "automated low power lidar, ALC"). Ceilometers are available as networks, e.g. in Europe. I don't know the situation in China but this option should be mentioned. It has been demonstrated that most ceilometers are capably to determine the MLH.

- L66: "only in the morning...". This statement is not true for all places in the world.

- L68: "the low vertical resolution": Most of the ceilometers (and lidars) provide a 15 m resolution that is fully sufficient to determine the CBL. In the framework of Ceilinex2015 we have compared instruments from Vaisala, Lufft, and Campbell. In particular for Lufft CHM15k and Vaisala CL51 no problems appeared (the corresponding AMT-paper was mainly on water vapor absorption corrections).

- L70: "low SBL height is not evaluated". I assume this is due to the overlap-problem. The reader is not aware of this at this point of the manuscript. So a short explanation should be added or the sentence should be moved to the next section.

- L85: Here the period of the measurements should be explicitly mentioned, and the time resolution, and the gaps in the measurement schedule if any. The coordinates of the location should be given with more decimal places.

- L89: "correction of overlap". As this is an essential point, it would be nice to read a few detail, at least the height of the lowest "useful" range bin should be given here (actually it is mentioned later, sometimes called "gate", sometimes "range").

- L94: To my knowledge Baars et al. (2008) were (one of) the first who applied the Haar-wavelet technique to continuous lidar measurements (de Haij et al. used the very old LD-40 with a limited vertical measurement range) including a detailed sensitivity analysis. This paper could be cited as well.

- L99: When mentioning "attenuated backscattering profile f(z)" here, this quantity should be defined previously. But in L89 only RCS and "then logarithm calculation" ($\ln(\beta'_{532})$) are mentioned, but not $f(z)$.

- L103: The interpretation of the wavelet approach may have some pitfalls: to associate the largest maximum of $W_f$ to the MLH (CBL) is obvious (though exceptions may occur), however, the interpretation of the first local maximum is critical (MLH'). Often the aerosols are structured and several internal layers appear making the allocation of a local maximum to an atmospheric feature very difficult. In the last years a lot of research has been devoted to estimate MLH resulting in a number of papers that should be mentioned here, e.g. Kotthaus et al., Geiß et al., Morille et al., de Bruine et al., Poltera et al. and many more.

- L113: It is stated that the temporal resolution of the MLH is one hour. In Figs. 1–4 the resolution seems to be better. Is this a inconsistency?

- L114: "eliminating ...false value and peak value": Please give a short hint, what is meant.

- L133: The description of the PMRS-model is quite relevant, so I suggest to move S2 to the main text. In particular the uncertainty of PM2.5 depending on the uncertainty of the MLH should be highlighted as the MLH and its (large) uncertainty is the main outcome of their study. Fig. 10 reveals extreme differences in case of the RS-retrieval.

- L139: What type of sunphotometer is used? Give a few comments on these measurements.

- Section 3.1: "case study" is mentioned in the caption, but it is not very clear, what this is. Four case studies each covering a period of 3 days corresponding to Figs. 1-4?

- Figs. 1–4: The triangles cover a vertical range of 300 m or so. What part of the symbol indicates the RS-retrieval? The top, the center, the bottom? In many cases the MLH seems to be zero.

- L151: "the aerosol layer height keep...": This sentence should be rephrased.

- L152: "MLH is always higher than MLH_RS,...": this seems to be in contradiction to the findings in L204. Please check all conclusions carefully.

- L175: "...RS is its very good precision". This should be explained. Why is the method conceptually superior to a ceilometer/lidar retrieval? The criteria involved for estimating the MLH from RS or lidars both have their "free parameters" (thresholds).

- L190: The histograms should be explained in much more detail to avoid confusion. The reader might expect that the columns (e.g. for winter) add up to 100 % (for the lidar and the RS retrieval). However, it seems that the total column is the annual relative frequency and the different colors indicate the contribution of each season to the total. If so, the seasonal distribution should be discussed as well. The overall agreement between the two data sets is actually low – neither the absolute values nor the shape of the distribution agree. Moreover, as stated by the authors, in 35 % of the cases no intercomparison is possible due to the overlap problem. This mainly occurs in spring – any idea why?

- Figs. 7 and 8: to compare these two figures (basically it is the same information, however, in Fig. 7 the annual mean and in Fig. 8 the seasonal means are shown?) relevant minimum/maximum values of the MLH should explicitly be given in the text. Then, it can be seen if the numbers are consistent. What is the "shaded area" in Fig. 7: from 550 m to 2000 m in case of MLH at 0000 hours? What are the consequences of such a large range for the significance of differences (in the course of the day, for inter-annual changes)?

- L228: According to the authors MLH' could be the residual layer or the stable layer (or any internal layer). So, the implications for the dispersion of pollutants are hard to infer. The benefit of MLH' should be clearly described in the paper (see comment above, and the conclusions of the manuscript).

- L236ff: The authors explain the "valley" in the diurnal cycle from the domination of the developing CBL over the RL (in terms of the signal gradient) after sunrise. In many publications the complete diurnal cycle is considered as the combination of the SL and the CBL. Then, a much smoother curve can be found. Moreover, the SL seems to be more relevant for the accumulation of pollutants close to the surface than the RL.

- L259: "...into near-surface air quality information". It has been shown by Geiß et al. (2017) that the near-surface air quality does not only depend on the MLH (see also comment L33, and manuscript L275).

- Fig. 9: Are the inter-annual differences of the annual cycles significant in view of the very large uncertainty ranges (see also comment above)?

- L262: A short description of the in-situ measurements should be added: where have they been made, what is their temporal resolution/coverage, and their accuracy. Is one site or an average over many sites considered? The differences between MLH_RS and the in-situ seem to be indeed too large for any air quality application.

- L297: Last sentence: the authors should be aware that many sophisticated methods to retrieve the diurnal cycle has been published recently. They should be cited (many references in the manuscript are quite old), see suggestions.

References mentioned above:

Baars, H., Ansmann, A., Engelmann, R., and Althausen, D.: Continuous monitoring of the boundary-layer top with lidar, Atmos. Chem. Phys., 8, 7281-7296, https://doi.org/10.5194/acp-8-7281-2008, 2008.

de Bruine, M., Apituley, A., Donovan, D. P., Klein Baltink, H., and de Haij, M. J.: Pathfinder: applying graph theory to consistent tracking of daytime mixed

layer height with backscatter lidar. Atmos. Meas. Tech. 10, 1893–1909, doi:10.5194/amt‐10‐1893‐2017, 2017.

Geiß, A., Wiegner, M., Bonn, B., Schäfer, K., Forkel, R., von Schneidemesser, E., Münkel, C., Chan, K. L., and Nothard, R.: Mixing layer height as an indicator for urban air quality?, Atmos. Meas. Tech., 10, 2969-2988, https://doi.org/10.5194/amt-10-2969-2017, 2017.

Kotthaus, S. and Grimmond, C. S. B.: Atmospheric Boundary Layer Characteristics from Ceilometer Measurements Part 1: A new method to track mixed layer height and classify clouds, Quart. J. RMetS, https://doi.org/10.1002/qj.3299, 2018a.

Morille, Y., Haeffelin, M., Drobinski, P., and Pelon, J.: STRAT: An automated algorithm to retrieve the vertical structure of the atmosphere from single-channel lidar data, JTech, 24(5), 761-775, https://doi.org/10.1175/JTECH2008.1, 2007.

Poltera, Y., Martucci, G., Collaud Coen, M., Hervo, M., Emmenegger, L., Henne, S., Brunner, D., and Haefele, A.: PathfinderTURB: an automatic boundary layer algorithm. Development, validation and application to study the impact on in situ measurements at the Jungfraujoch. Atmos. Chem. Phys. 2017, 17, 10051-10070, doi:10.5194/acp-17-10051-2017.

---

## Referee Comment (RC2) · Anonymous Referee #2 · 15 May 2020

This manuscript reports climatological values of mixing layer height (MLH) over Beijing for from Lidar measurements, their evaluation against radiosonde based MLH calculation and argues that MLH values derived from Lidar is better than that from RS for estimation of PM2.5 at surface. The manuscript needs a lot of improvement before publication. Please see comments and suggestions below:

1) The first part of the results and discussion, where the comparison of the various approaches to estimate MLH is presented is not satisfactory. First, the English language is poorly written which makes it hard to understand what is being conveyed. Also, the

text corresponding to Figures 2-4 lacks any discussion of the features seen in them. More detailed discussion and language improvement is needed.

2) Fig 3d: why is MLH_RS detected at 0.6 km? it should be $\sim$ 2km as seen in Fig 3c.

3) Actually, the whole context of the first 4 figures is not understood. Why are these days shown here? why not any other day? Is it meant to show seasonal variability i.e one for one season? If the figures are introduced to show various types of differences between the 3 methods, then the discussion show be organized in that manner and the inter comparison figure 5 should be discussed in context of features seen in these figures 1-4. Better organization is needed.

4) Please give more details about the cloud screen/flag used (Line 111). How was cloud detected, at what resolution etc?

5) Fig 1a, 2a,3a,4a all are stretched, the fonts are of different sizes compared to other panels and should be made consistent. Figure 1b have an undefined variable MLH2 in the figure label?

6) Evaluation of MLH_lidar is essential for second half of this manuscript. Hence, more detailed comparison and discussion is necessary. scatter plots in supplementary should be presented in main and discussed in detail. Moreover, as seasonal variability is significant on MLH, the comparison in Figure 6 should include and discuss scatter plots for all the seasons, separately.

7) From Fig 1-5, MLH from Lidar is termed as MLH but in Fig 6, it is termed as MLH_Lidar. Please be consistent.

8) Fig 9: Is the MLH from RS also showing yearly variability similar to the MLH derived from Lidar? Please include the same from RS also in this figure and discuss.

9) In Figure 10d shows that correlations are highest when RMSE is also highest, this is non intuitive, please describe this feature. Also, the comparison in Figure 10 should be presented separately for each season as MLH has strong seasonal variations.

10) Throughout the manuscript standard deviation and RMSE is used interchangeable, which should be corrected. They are not same.

11) Please check carefully for the typos in the manuscript. The title itself has one "airpollution".

---

## Author Comment (AC2) · 19 May 2020

**Reply to Matthias Wiegner:**

We appreciate the time and effort you spent on this manuscript for the very detailed and helpful review, which gives me a lot of inspiration and guide. We have carefully revised our manuscript. The point-by-point answers are as follows.

**Comments from the editors and reviewers:**
**General comments:**

For the time being there is one reviewer's comment available. I agree with his/her general comments and suggestions so it is not necessary to repeat their remarks. Below please find a few additional comments focusing on selected technical or scientific details. As I am not an assigned reviewer I have only focused on the air quality aspect and the meaning of MLH'. Anyway, I hope that they can contribute to improve the paper.

1. Title: Mentioning "implications for air pollution" in the manuscript's title is a little bit misleading. The authors describe a methodology to derive the mixing layer height (convective, stable, residual layer (CBL, SL, RL)) from lidar measurements. They demonstrate that the agreement with radiosonde based retrievals is often quite limited. MLH is used as input for a numerical model (i.e. one equation) to estimate ground-based PM2.5 concentrations. The agreement with in-situ measurements is also limited. Consequently the whole procedure results in a rough estimate of PM2.5 only. Having this uncertainty and the heterogeneity of local sources of pollutants in mind (see also comments below), I feel that mentioning it in the title of the paper is somewhat exaggerated.

   **A1**: The content is added to try to response to the comment. "The vertical structure of the mixing layer is important for the concentrations at the surface due to its impact on the volume into which pollutants are mixed. Mues et al., (2017) reported that black carbon concentrations show a clear anticorrelation with MLH measurements. Hu et al., (2014) found an anticorrelation between near-surface $O_3$ and MLH for seven cities in the North China Plain. In the study, as shown in Fig. 11, the correlation between $MLH_L$ and observed $PM_{2.5}$ data from the same observatory shows high negative correlation (R=-0.569) with the four years measurement (2014-2017). Actually, the pollutant concentration near surface is affected by the overall effect of the local emission and meteorological condition, with variation of different spatio-temporal distribution. MLH is just one of these influencing factor. Geiß et al., (2017) indicates that when MLH and near-surface concentrations are linked, it is necessary to take the locations, i.e., meteorological conditions and local sources, and the details of the MLH retrieval into account. In fact, all the data used in our study is observed from the same observatory, and PMRS model used to calculate the surface $PM_{2.5}$ concentration includes the parameters of the emission (AOD) and meteorological condition (RH) into account."

As to the agreement with the in-situ, "Considering the uncertainty of the group of parameters used in the model, the agreement between calculated $PM_{2.5}$_lidar and in-situ measurement is reasonable good."

2. L28: "...identification of pollutant emissions and sources": This statement seems to be misleading. The MLH (among others) has an influence of the dispersion of pollutants, but how sources can be identified from the MLH is not obvious and must be explained if the authors insist on that statement.

    **A2**: The sentence is rephrased by "quantification of pollutant emissions (Haeffelin et al., 2012; Seibert et al., 2000; Baars et al., 2008; Liu and Liang, 2010; Bruine et al., 2017)." With the knowledge of MLH and the surface pollutant concentration, the total pollutant emissions can be calculated when it is well mixed.

3. L33: "contributes to the assessment of the pollutant concentration near the surface": I agree that MLH "contributes" to concentrations of pollutants but it must be kept in mind that the spatio-temporal distribution of sources plays a dominating role; see Geiß et al., (2017).

    **A3:** Yes, you are right. The pollutant concentration near surface is affected by the overall effect of the local emission and meteorological, with variation of different spatio-temporal distribution. MLH is just one of these influencing factor. In fact, all the data used in our study is observed from the same observatory, and the surface pollutant model used take the emission (AOD) and meteorology (RH) into account.

4. L38: "MLH can be estimated by ... the concentration of PBL constituents". It is rather the other way around (having in mind the inherent problems already mentioned).

    **A4:** The sentence is rephrased as "MLH can be estimated by the measurement of variance of the mechanical turbulence, of the temperature enabling convection or of the substance content in the low troposphere."

5. L59: Here ceilometers should be explicitly mentioned. Lidars are comparably rare instruments, and only a few can be operated unattendedly and fully automatic (MPL is one of them, so often called a ceilometer or an "automated low power lidar, ALC"). Ceilometers are available as networks, e.g. in Europe. I don't know the situation in China but this option should be mentioned. It has been demonstrated that most ceilometers are capably to determine the MLH.

    **A5:** The comment is added as "With recent upgrades of the hardware, ceilometer, an known as automated low power lidar, or automated lidars and ceilometers, has been demonstrated that be capably to determine the MLH."

6. L66: "only in the morning...". This statement is not true for all places in the world.

    **A6:** Yes, the statement is not all true, taking into consideration that additional RS may be launched in other time. 08:00 LT and20:00 LT is the universal launch time (Collaud et al., 2014; Guo et al.,

2016; Su et al., 2019). The sentence has been rephrased as "The meteorological radiosondes usually acquire the MLH in the morning (08:00 LT) and at night (20:00 LT), when the diurnal cycle of ML combined with stable and convective PBL cannot be well characterized."

7. L68: "the low vertical resolution": Most of the ceilometers (and lidars) provide a 15 m resolution that is fully sufficient to determine the CBL. In the framework of Ceilinex2015 we have compared instruments from Vaisala, Lufft, and Campbell. In particular for Lufft CHM15k and Vaisala CL51 no problems appeared (the corresponding AMT-paper was mainly on water vapor absorption corrections).

   **A7:** The description is removed.

8. L70: "low SBL height is not evaluated". I assume this is due to the overlap problem. The reader is not aware of this at this point of the manuscript. So a short explanation should be added or the sentence should be moved to the next section.

   **A8:**  It is moved the next section.

9. L85: Here the period of the measurements should be explicitly mentioned, and the time resolution, and the gaps in the measurement schedule if any. The coordinates of the location should be given with more decimal places.

   **A9:** The corresponding content has been added "The period of the measurements is from 2013 to 2018, nearly six years. Except for the MLH data from lidar of 2013 mainly existing in winter and spring, the measurement of 2014-2018 are all annual continued observations."" CE370 can detect a long range profile up to 30 km every 1 second. For the enhancement of signal noise ratio, 60 profiles are averaged to restore as one with the time resolution of 1 minute." "the observation site (116.379° E, 40.005° N) in Beijing city"

10. L89: "correction of overlap". As this is an essential point, it would be nice to read a few detail, at least the height of the lowest "useful" range bin should be given here (actually it is mentioned later, sometimes called "gate", sometimes "range").

    **A10:** The descriptions of overlap is present in the revised manuscript as "Due to the design of the lidar, the received view close to the ground does not completely coincide with transmitted view. There exist a detection blind area of lidar and a geometric overlap factor is used to correct the mismatch of field of view." and "Due to the limitation of algorithm and insufficient lidar overlap, the minimum range of the MLH calculation is on the order of 250 m."

11. L94: To my knowledge Baars et al. (2008) were (one of) the first who applied the Haar-wavelet technique to continuous lidar measurements (de Haij et al. used the very old LD-40 with a limited vertical measurement range) including a detailed sensitivity analysis. This paper could be cited as well.

    **A11**: It has been added.

12. L99: When mentioning "attenuated backscattering profile f(z)" here, this quantity should be defined previously. But in L89 only RCS and "then logarithm calculation" ($\ln(\beta'_{532})$) are mentioned, but not f(z).

**A12**: It is added as "RCS is also expressed as f (z), with z the measurement height."

13. L103: The interpretation of the wavelet approach may have some pitfalls: to associate the largest maximum of Wf to the MLH (CBL) is obvious (though exceptions may occur), however, the interpretation of the first local maximum is critical (MLH'). Often the aerosols are structured and several internal layers appear making the allocation of a local maximum to an atmospheric feature very difficult. In the last years a lot of research has been devoted to estimate MLH resulting in a number of papers that should be mentioned here, e.g. Kotthaus et al., Geiß et al., Morille et al., de Bruine et al., Poltera et al. and many more.

**A13**: It is added as "Actually, every local maximum corresponds an aerosol layer and several internal layers appear making the allocation of a local maximum to an atmospheric feature very difficult (Morille et al., 2008; Geiß et al., 2017; Poltera et al., 2017; Kotthaus et al., 2018)." And "However, the interpretation of the first local maximum (MLH') is critical. To form a diurnal cycle of MLH from these several layers, a geodesic approach was applied to pathfinderTURB (Poltera et al., 2017), while COBOLT (Geiß et al., 2017) uses a time–height-tracking approach with moving windows. Nevertheless, these method all are based on the selection of the lowest detected aerosol. The height of the lowest detected aerosol layer was regarded as the daytime MLH and the nocturnal stable boundary layer, respectively, as reported by Mues et al. (2017) and Kotthaus et al. (2018). Su et al. (2019) developed a DTDS algorithm, started with the lowest point and tracked depending time and stability, but the nocturnal MLH is not evaluated. Detection of nocturnal boundary-layer heights, in contrast to the residual layer, is a major challenge (Haeffelin et al., 2012; Lotteraner and Piringer, 2016; de Bruine et al., 2017). SBL seems to be more relevant for the accumulation of pollutants close to the surface than the RL in the evening and early morning. Thus, one of the objective of this study is to investigate the usefulness of $MLH_L$' from CE-370 to capture the SBL height over Beijing."

14. L113: It is stated that the temporal resolution of the MLH is one hour. In Figs. 1–4 the resolution seems to be better. Is this a inconsistency?

**A14:** You are right. The resolution of Figs. 1-4 is one minute, according to the primary MLH_lidar result. The description of "with the time resolution of 1 minute. For convenient comparison with air quality and meteorological parameters, all MLH results are one hour averaged" is added in the manuscript.

15. L114: "eliminating ...false value and peak value": Please give a short hint, what is meant. L133: The description of the PMRS-model is quite relevant, so I suggest to move S2 to the main text. In particular the uncertainty of PM2.5 depending on the uncertainty of the MLH should be highlighted as the MLH and its (large) uncertainty is the main outcome of their study. Fig. 10 reveals extreme differences in case of the RS-retrieval.

**A15:** Due to incomplete screen of cloud and rainfall, it can be misjudged as the MLH, leading to some false and very large value, which should be eliminated. Now the sentence has been rephrased by "unrealistic outliers are deleted"

RMRS-model is moved to the main text.

"The correlations at 12, 13, 14, and 15 LST were 0.894, 0.922, 0.927, and 0.900, respectively. The higher accuracy may be due to the completed mixing of the aerosol at noon and the vertical distribution of the aerosol tend to be uniform. The correlation between 8, 9 and 17 LST is less than 0.8, and the relatively poor accuracy is related to the complex boundary layer structure in the morning and evening. It is difficult to achieve fully mixing of the aerosol in the stable boundary layer or the residual layer. The daily variation of calculated surface pollutant accuracy using MLH retrieval by lidar vary with the daily variation of aerosol mixing uniformity at different times during the daytime."

The calculated PM2.5_RS from $MLH_{RS}$ and in-situ measurement shows great discrepancy, as the reason that "above $MLH_{RS}$ there still exist a large amount of aerosol. The discrepancy makes sense using the method with the determinate total amount of pollutant of the column atmosphere. The gap may be narrowed if the total emission from surface is used."

16. L139: What type of sunphotometer is used? Give a few comments on these measurements.

**A16:** The comment is added as "All the parameter is observed by the instruments employed in the same observatory of lidar. The optical parameters of the column aerosols (AOD and FMF.) are obtained by a sky-sun photometer (CE318-DP, CIMEL, France), which is affiliated with the Aerosol RObotic NETwork (AERONET) (Holben et al., 1998; Dubovik, 2000). Measurements are automatically scheduled with direct sun irradiance measurements each of about 15 min and angular sky radiance scanning of about 1 h each (Li et al., 2015; Che et al., 2014; Wang et al., 2019). Atmospheric meteorological data (relative humidity-RH, wind speed-WS, wind direction-WD, etc.) are obtained by automatic meteorological monitoring station (BLJW-4). $PM_{2.5}$ mass concentration is obtained by $PM_{2.5}$ monitor (BAM-1020, MetOne, USA), which shows good agreement with the measurement of national monitoring network near the observatory. All the data is quality controlled and calculated as one hour averaged and the measurement period is from 2014 to 2018."

17. Section 3.1: "case study" is mentioned in the caption, but it is not very clear, what this is. Four case studies each covering a period of 3 days corresponding to Figs.1-4?

**A17:** "case study" in the caption is removed in the revised manuscript.

18. Figs. 1–4: The triangles cover a vertical range of 300 m or so. What part of the symbol indicates the RS-retrieval? The top, the center, the bottom? In many cases the MLH seems to be zero.

**A18:** The top of the triangle indicats the RS-retrieval.

19. L151: "the aerosol layer height keep...": This sentence should be rephrased.

    **A19:** It has been rephrased.

20. L152: "MLH is always higher than MLH_RS,...": this seems to be in contradiction to the findings in L204. Please check all conclusions carefully.

    **A20:** The sentence of "MLH is always higher than MLH_RS,..." is removed, while "MLH_RS tends to be larger than MLH" is rephrased as "MLH_RS tends to be larger than MLH in the afternoon". Actually, we want to express that MLH tend to be higher than MLH_RS in the evening and early morning, while MLH is lower than MLH_RS in the afternoon.

21. L175: "...RS is its very good precision". This should be explained. Why is the method conceptually superior to a ceilometer/lidar retrieval? The criteria involved for estimating the MLH from RS or lidars both have their "free parameters" (thresholds).

    **A21:** The paper focus the measurement of lidar. Indeed, it is improper to emphasize the precision of RS. Yet, from my perspective ,in view of the definition of MLH inferring "convection or mechanical turbulence" (COST action 710 – Final report, 1998), RS measurement is likely to close to the two parameter, "due to their ability to characterize the thermodynamic and dynamic states of the boundary layer". And, a lot of study has been done to evaluate the precision of remote sensing measurement, with the comparison with RS (Wiegner et al., 2006; Milroy et al., 2012; Sawyer and Li, 2013; Cimini et al., 2013; Tang et al, 2016; Singh et al., 2016; Mues et al., 2017; Su et al, 2019). And, in section 2.3, there is the statement of "In most cases, the exact threshold value has only a small impact on the PBL height due to the large slope of $Ri_b$ in this interval (Collaud et al., 2014)."

    Nevertheless, the sentence has been rephrase as "The temporal resolution (usually two or three measurements per day) of PBL detection by RS is not able to provide the mixing layer height diurnal cycle, no matter its good precision."

22. L190: The histograms should be explained in much more detail to avoid confusion. The reader might expect that the columns (e.g. for winter) add up to 100 % (for the lidar and the RS retrieval). However, it seems that the total column is the annual relative frequency and the different colors indicate the contribution of each season to the total. If so, the seasonal distribution should be discussed as well. The overall agreement between the two data sets is actually low – neither the absolute values nor the shape of the distribution agree. Moreover, as stated by the authors, in 35 % of the cases no intercomparison is possible due to the overlap problem. This mainly occurs in spring – any idea why?

    **A22:** Yes, It is added that "the total column is the annual relative frequency and the different colors indicate the contribution of each season to the total." The description of Fig. 5 is rephrased that "Fig. 5. Comparison of frequency distribution of all MLH (2013-2018) retrieved from lidar and radiosonde with the supplementary information of seasonal variation. Noted that for presenting the detail distribution, $MLH_L$ adds up to 20%, while $MLH_{RS}$ add up to 45%."

Actually, the seasonal distribution was discussed in the manuscript, as "As to the seasonal variation of both lidar and RS measurement at 0800 LST, the frequency of larger $MLH_L$ value in summer is minimal, indicating summer MLH is lower than other season. As for radiosonde, $MLH_L$ lower than 0.25 km mostly distributes in winter, with the rate of around 15% for both 0800 and 2000 LST, and the frequency decreases rapidly when $MLH_L$ gets larger than 0.25 km."

It is added that "This lower values mainly occurs in winter and autumn, when it tends to present lower MLH (Tang et al., 2016)."

23. Figs. 7 and 8: to compare these two figures (basically it is the same information, however, in Fig. 7 the annual mean and in Fig. 8 the seasonal means are shown?) relevant minimum/maximum values of the MLH should explicitly be given in the text. Then, it can be seen if the numbers are consistent. What is the "shaded area" in Fig. 7: from 550 m to 2000 m in case of MLH at 0000 hours? What are the consequences of such a large range for the significance of differences (in the course of the day, for inter-annual changes)?

**A23:** Besides the annual mean diurnal cycle, Fig. 7 also compared the mean MLH with MLH and MLH_RS, revealing that MLH shows overall good CBL height and MLH' generally cannot indicate SBL well. In the revised manuscript, Fig. 7 has been remove to the section of comaparisong of different MLH approach. The seasonal and annual variation is presented mainly based on MLH.

The content and table is added. "The maximum in summer is 1.526 km, and the maximum in autumn is 1.445 km. From the average of the four seasons, the averages in spring, summer, autumn and winter are 1.409 km, 1.261 km, 1.297 km and 1.228 km, respectively. The average value in autumn (1.297km) is greater than that in summer (1.261km)."

Table 1 Statistics of boundary layer height seasonal change

| MLH/km | Spring | summer | autumn | winter |
|---|---|---|---|---|
| mean | 1.409 | 1.261 | 1.297 | 1.228 |
| maximum | 1.647 | 1.526 | 1.445 | 1.404 |
| minimum | 1.126 | 0.932 | 1.117 | 1.098 |

The "shaded area" in Fig. 7 indicates the standard deviation of MLH and MLH'. In statistics, mean ± standard deviation is commonly used to indicate the average and degree of dispersion of a set of data, referred Su et al. (2019) and Geiß et al., (2017). The inter-annual changes is analyzed based on the mean result, even if the data dispersion is large, it still has statistical significance.

Su, T., Z. Li, and R. Kahn, 2019: A new method to retrieve the diurnal variability of planetary boundary layer height from lidar under different thermodynamic stability conditions, Remote Sens. Environ., 237, doi:10.1016/j.rse.2019.111519.

24. L228: According to the authors MLH' could be the residual layer or the stable layer (or any internal layer). So, the implications for the dispersion of pollutants are hard to infer. The benefit of MLH'

should be clearly described in the paper (see comment above, and the conclusions of the manuscript).

**A24:** It is added in the conclusions that "MLH$_L$' have the potential to describe the stable layer height at night sometime, even though the capability is limited due to the high incomplete overlap of CE-370. The stable layer height detected by MLH$_L$' in the nighttime is the layer in which ground-emitted atmospheric pollutants are trapped, it contributes to the assessment of the surface pollutant concentration when there is emission in the nocturnal time using the numerical models."

25. L236: The authors explain the "valley" in the diurnal cycle from the domination of the developing CBL over the RL (in terms of the signal gradient) after sunrise. In many publications the complete diurnal cycle is considered as the combination of the SL and the CBL. Then, a much smoother curve can be found. Moreover, the SL seems to be more relevant for the accumulation of pollutants close to the surface than the RL.

**A25:** The application of height of SL and RL may be different, as discussed in the conclusion section that "The stable layer height detected by MLH$_L$' in the nighttime is the layer in which ground-emitted atmospheric pollutants are trapped, it contributes to the assessment of the surface pollutant concentration when there is emission in the nocturnal time using the numerical models. Whilst the residual layer height corresponding to trapped atmospheric constituents discharged some hours before, which can be employed to convert column-mean optical depths into near-surface air quality information from remote sensing."

26. L259: "...into near-surface air quality information". It has been shown by Geiß et al. (2017) that the near-surface air quality does not only depend on the MLH (see also comment L33, and manuscript L275).

**A26**: Please see the response to the comment 3.

27. Fig. 9: Are the inter-annual differences of the annual cycles significant in view of the very large uncertainty ranges (see also comment above)?

**A27:** The inter-annual changes is analyzed based on the mean result, even if the data dispersion is large, it still has statistical significance.

28. L262: A short description of the in-situ measurements should be added: where have they been made, what is their temporal resolution/coverage, and their accuracy. Is one site or an average over many sites considered? The differences between MLH_RS and the in-situ seem to be indeed too large for any air quality application.

**A28:** It is added that "All the parameter is observed by the instruments employed in the same observatory of lidar. The optical parameters of the column aerosols (AOD and FMF.) are obtained by sky-sun photometer (CE318-DP, CIMEL, France), which is affiliated with the Aerosol RObotic NETwork (AERONET) (Holben et al., 1998; Dubovik, 2000). Measurements are automatically

scheduled with direct sun irradiance measurements each of about 15 min and angular sky radiance scanning of about 1 h each (Li et al., 2015; Che et al., 2014; Wang et al., 2019). Atmospheric meteorological data (relative humidity-RH, wind speed-WS, wind direction-WD, etc.) are obtained by automatic meteorological monitoring station (BLJW-4). PM2.5 mass concentration is obtained by PM2.5 monitor (BAM-1020, MetOne, USA), which shows good agreement with the measurement of national monitoring network near the observatory. All the data is quality controlled and calculated as one hour averaged and the measurement period is from 2014 to 2018."

Like to the response of comment 15, the calculated PM2.5_RS from MLH$_{RS}$ and in-situ measurement shows great discrepancy, as the reason that "above MLH$_{RS}$ there still exist a large amount of aerosol. The discrepancy makes sense using the method with the determinate total amount of pollutant of the column atmosphere. The gap may be narrow if the total emission from surface is used."

29. L297: Last sentence: the authors should be aware that many sophisticated methods to retrieve the diurnal cycle has been published recently. They should be cited (many references in the manuscript are quite old), see suggestions.

**A29:** The reference of "Wiegner et al., 2006; de Bruine et al., 2017; Morille et al., 2017; Kotthaus et al., 2018" is cited.

The reference is added in the revised manuscript.

Baars, H., Ansmann, A., Engelmann, R., and Althausen, D.: Continuous monitoring of the boundary-layer top with lidar, Atmos. Chem. Phys., 8, 7281-7296, https://doi.org/10.5194/acp-8-7281-2008, 2008.

de Bruine, M., Apituley, A., Donovan, D. P., Klein Baltink, H., and de Haij, M. J.: Pathfinder: applying graph theory to consistent tracking of daytime mixed layer height with backscatter lidar. Atmos. Meas. Tech. 10, 1893–1909, doi:10.5194/amt-10-1893-2017.

Geiß, A., Wiegner, M., Bonn, B., Schäfer, K., Forkel, R., von Schneidemesser, E., Münkel, C., Chan, K. L., and Nothard, R.: Mixing layer height as an indicator for urban air quality?, Atmos. Meas. Tech., 10, 2969-2988, https://doi.org/10.5194/amt-10-2969-2017, 2017.

Kotthaus, S. and Grimmond, C. S. B.: Atmospheric Boundary Layer Characteristics from Ceilometer Measurements Part 1: A new method to track mixed layer height and classify clouds, Quart. J. RMetS, https://doi.org/10.1002/qj.3299, 2018.

Morille, Y., Haeffelin, M., Drobinski, P., and Pelon, J.: STRAT: An automated algorithm to retrieve the vertical structure of the atmosphere from single-channel lidar data, JTech, 24(5), 761-775, https://doi.org/10.1175/JTECH2008.1, 2007.

Poltera, Y., Martucci, G., Collaud Coen, M., Hervo, M., Emmenegger, L., Henne, S., Brunner, D., and Haefele, A.: PathfinderTURB: an automatic boundary layer algorithm. Development, validation and application to study the impact on in situ measurements at the Jungfraujoch. Atmos. Chem. Phys. 2017, 17, 10051-10070, doi:10.5194/acp-17-10051-2017.

---

## Author Response (AR1)

**Response to referees on "Determination and climatology of diurnal cycle of atmospheric mixing layer height over Beijing 2013-2018: Lidar measurements and implication for air pollution" by Haofei Wang et al.**

Haofei Wang[1,2], Zhengqiang Li[1]*, Yang Lv[1,2], Ying Zhang[1], Hua Xu[1], Jianping Guo[3], Philippe Goloub[4]

1. State Environmental Protection Key Laboratory of Satellite Remote Sensing, Aerospace Information Research Institute, Chinese Academy of Sciences, Beijing, 100101, China
2. University of Chinese Academy of Sciences, Beijing, 100101, China
3. State Key Laboratory of Severe Weather, Chinese Academy of Meteorological Sciences, Beijing, 100081, China
4. Laboratoire d'Optique Atmospherique, UMR8518, CNRS – Universit´e de Lille 1, Villeneuve d'Ascq, Lille, 59000, France

We appreciate the reviewers' comments on the manuscript. All comments are highly valuable and helpful for us to improve our manuscript. We have studied them carefully and have addressed them in the revised manuscript, which includes additional investigations. The modification corresponding to Referee #1 is marked in red color, while that responding to Matthias Wiegner and Referee #2 are shown in blue color. The content present in green color is rephrased according to the similarity report. Below are point-by point responses to the referee's comments. The referees' comments below are marked in grey, followed by authors' response in black.

**Reply to Anonymous Referee #1:**

**General comments:**

This manuscript analyses long-term measurements of mixed layer height (MLH) over Beijing. Authors describe and evaluate the techniques, derive climatological diurnal variation, and presents an application towards the estimation of fine particulate matter. Several comments and suggestions are offered for authors to consider while revising the manuscript for ACP.

**Major points:**

1. Introduction section should provide more background, based on studies comparing mixed layer measurements, not limited to LIDAR but also from RADAR and other instruments. Some discussion has been made on the importance of MLH in context of air pollution mixing and dispersion, which should be corroborated with relevant recent references (e.g. Singh et al., 2016; Mues et al., 2017).

   **A1**: Thank you to mention the question from this perspective. The measurement of RADAR, microwave radiometer, ceilometer is added in the revised manuscript. "Singh et al.(2016) investigate the evolution of the Local Boundary Layer in the central Himalayan region, using a radar wind profiler detecting wind components based on signal to-noise ratio profile. Collaud et al. (2014) compared the MLH measurement of microwave radiometer from atmospheric

temperature profile with other measurement in Swiss plateau. Mues et al. (2017) used the ceilometer to retrieve the MLH based on aerosol backscatter signal in the Kathmandu Valley." **(P18L42)** "Recent studies compared remote sensing measurements (lidar, radar wind profiler, microwave radiometer) with radiosonde (RS) (Milroy et al., 2012; Sawyer and Li, 2013; Cimini et al., 2013; Tang et al, 2016; Singh et al., 2016; Mues et al., 2017; Su et al, 2019)"**(P18L70)**.The relevant recent references (Singh et al., 2016; Mues et al., 2017) is added when talk about air pollution mixing and dispersion in the first paragragh.**(P18L37)**

2. Stronger correlations between LIDAR and Radiosonde are seen during afternoon but such correlations are absent in morning and evening. Besides poor correlation, values of MLH also do not match with radiosonde in morning and evening. There should be more deeper analysis and discussions on these aspects with references to previous studies.

   **A2**: It is necessary. Discussions is added as "The poor agreement between MLH (MLH') from lidar and MLH_RS is also reported in the study of Su et al. (2019), in which shows that the correlation of PBLH measurement between lidar and radiosonde is 0.14 at 0630 LST. The significant scatter in the morning and evening is associated with complicated structure of boundary layer, as indicated by the existence of stable boundary layer and residual layer (Su et al., 2019; Tang et al., 2016). In this study, more than 35% measurement of SBL height is not within the scope of the lidar detection. Additionally, under stable conditions, it is difficult to estimate the MHL from lidar data in some cases, due to the weak vertical gradients in the aerosol content. In the evening and early morning, problems arise from finding a sufficiently clear change in the backscatter profile at the top of the SBL, within the previously well-mixed layer (Russell et al., 1974; Seibert et al., 2000)." **(P26L253)**

3. The correlation analysis between MLH' and radiosonde should be shown for all three times in the supplement.

   **A3**: It is added as Fig. S4 in the supplementary.

[Figure]

[Figure]

4. Section 3.1 describes mostly the variations as retrieved with limited new insights into boundary layer evolution. Additionally, several general statements are made e.g. "MLH' sometime agree well with SBL". Remove general statements and provide more specific discussions based on analysis.

   **A4**: General statements is removed from Section 3.1, and add some description of boundary layer evolution, as well as some specific discussions based on analysis, see the revised manuscript.

5. Figure 8 shows significant reduction in MLH after sunrise, particularly during summer (l.245). This should be elaborated. How do the horizontal winds change during this time of minimum MLH? Have any other studies reported such variability in Beijing or elsewhere?

   **A5**: Deep analysis is added, as well as the study in Beijing of Tang et al. (2016). "It should be noted that summer exists the biggest amplitude of diurnal variation, with the deepest valley (0.93 km) increasing to the peak value of 1.51 km. Tang et al. (2016) indicate that the lower MLH value for summer nights and early mornings is contributed to the effect of the mountain plain wind. Beijing is located in the North China Plain, with Taihang Mountain in the west and Yanshan Mountain in the north. When the local mountain breeze from the northeast in the summer night superimposes the surface cooling, leading to the increase the thickness of the inversion layer, the height of the mixed layer gradually decreases. After sunrise, with the drive of thermal turbulence, the residual layer height observed by lidar is gradually replaced by a convective boundary layer height, with MLH increasing rapidly, and after 12:00 LT, the plain wind from the south-westerly direction gradually dominates."**(P28L318)**

6. Interannual variation – It seems that some of the years have data limited to particular season (s). it will be appropriate to compare the years which have consistency in the seasonal coverage. Otherwise better to analyze a particular season among different years. In any case it is not clear what new is learnt by this analyses. There should be supporting analyses of temperature /winds and /or aerosol changes to explain observed inter-annual variations.

**A6**: Except for the MLH data from lidar of 2013 mainly existing in winter and spring, the measurement of 2014-2018 are all annual continued observations, covering all the seasons **(P20L110)**. From 2014 to 2018, the magnitude of diurnal cycle of MLH increase year by year, indicating the volume available for the dispersion of pollutants extend, which is beneficial to the mitigation of surface pollution**(P28L337)**. Correlation of MLH and wind speed, relative humidity, temperature and AOD are calculated **(P29L347)**. The observed inter-annual variations can be explained by the aerosol change, the detail please see the manuscript **(P29L353)**. The statistic of interannual variation of the average, maximum and minimum of MLH, as well as CBL, is presented in the revised manuscript. **(P28334)**

7. There are several variables defined MLH, MLH', MLH_RS etc. Try to use these variables-consistently. For example in discussions MLH_RS is used but in Fig S3 -axis title -it is written as MLH.

   **A7**: Thank you for mentioning this question. It is easy to be confused. In the revised manuscript, "$MLH_L$" indicates the biggest local maximum from lidar, and "$MLH_L'$" indicates the first local maximum from lidar, and "$MLH_{RS}$" presents MLH from radiosonde. All the content of the manuscript and figure has been revised.

[Figure]

8. Correlations are very weak r = 0.01 between radiosonde and lidar at 8LST (Fig S4) and data has significant scatter. Add some inter-comparison studies from literature to elaborate on this.

   **A8:** As far as we can see, this comment is similar as comment 2. Please infer the author comment of comment 2.

9. Computation of PM2.5 should be part of the main manuscript, instead of supplementary material.

   **A9:** Computation of PM2.5 is transferred from supplementary to the main manuscript. **(P23L177)**

10. Fig 7: show variability in RS data too.

**A10:** The variability is added as standard deviation in Fig. 7 below.

[Figure]

**Minor changes**

11. Manuscript needs careful proofreading for language as various places. e.g. l.69: change consistent"
to "consistency" l.107: "MHL" to "MLH" l.118: check the sentence: " ..where was located",
probably it should be "which was located" l.195: change "collapse" to "collapsed" and "develop"
to "developed" l.203: "shown" to "shows"

**A11:** All the cacography mentioned is corrected, as well as other spelling errors.

**Reply to Matthias Wiegner:**

**General comments:**

For the time being there is one reviewer's comment available. I agree with his/her general comments
and suggestions so it is not necessary to repeat their remarks. Below please find a few additional
comments focusing on selected technical or scientific details. As I am not an assigned reviewer I have
only focused on the air quality aspect and the meaning of MLH'. Anyway, I hope that they can
contribute to improve the paper.

1. Title: Mentioning "implications for air pollution" in the manuscript's title is a little bit
misleading. The authors describe a methodology to derive the mixing layer height (convective,
stable, residual layer (CBL, SL, RL)) from lidar measurements. They demonstrate that the
agreement with radiosonde based retrievals is often quite limited. MLH is used as input for a
numerical model (i.e. one equation) to estimate ground-based PM2.5 concentrations. The
agreement with in-situ measurements is also limited. Consequently the whole procedure

results in a rough estimate of PM2.5 only. Having this uncertainty and the heterogeneity of local sources of pollutants in mind (see also comments below), I feel that mentioning it in the title of the paper is somewhat exaggerated.

**A1**: The content is added to try to response to the comment. "The vertical structure of the mixing layer is important for the concentrations at the surface due to its impact on the volume into which pollutants are mixed. Mues et al., (2017) reported that black carbon concentrations show a clear anticorrelation with MLH measurements. Hu et al., (2014) found an anticorrelation between near-surface $O_3$ and MLH for seven cities in the North China Plain. In the study, as shown in Fig. 11, the correlation between $MLH_L$ and observed $PM_{2.5}$ data from the same observatory shows high negative correlation (R=-0.569) with the four years measurement (2014-2017). Actually, the pollutant concentration near surface is affected by the overall effect of the local emission and meteorological condition, with variation of different spatio-temporal distribution. MLH is just one of these influencing factor. Geiß et al., (2017) indicates that when MLH and near-surface concentrations are linked, it is necessary to take the locations, i.e., meteorological conditions and local sources, and the details of the MLH retrieval into account. In fact, all the data used in our study is observed from the same observatory, and PMRS model used to calculate the surface $PM_{2.5}$ concentration includes the parameters of the emission (AOD) and meteorological condition (RH) into account."**(P29L361)**

As to the agreement with the in-situ, "Considering the uncertainty of the series of parameters used in the model, the agreement between calculated $PM_{2.5}$_lidar and in-situ measurement is reasonable good."**(P30L384)**

2. L28: "...identification of pollutant emissions and sources": This statement seems to be misleading. The MLH (among others) has an influence of the dispersion of pollutants, but how sources can be identified from the MLH is not obvious and must be explained if the authors insist on that statement.

**A2**: The sentence is rephrased by "quantification of pollutant emissions (Haeffelin et al., 2012; Seibert et al., 2000; Baars et al., 2008; Liu and Liang, 2010; Bruine et al., 2017)." **(P18L32)** With the knowledge of MLH and the surface pollutant concentration, the total pollutant emissions can be calculated when it is well mixed.

3. L33: "contributes to the assessment of the pollutant concentration near the surface": I agree that MLH "contributes" to concentrations of pollutants but it must be kept in mind that the spatio-temporal distribution of sources plays a dominating role; see Geiß et al., (2017).

**A3:** Yes, you are right. The pollutant concentration near surface is affected by the overall effect of the local emission and meteorological, with variation of different spatio-temporal distribution. MLH is just one of these influencing factor. In fact, all the data used in our study is observed from the same observatory, and the surface pollutant model used take the emission (AOD) and meteorology (RH) into account. **(P29L366)**

4. L38: "MLH can be estimated by ... the concentration of PBL constituents". It is rather the other way around (having in mind the inherent problems already mentioned).

**A4:** The sentence is rephrased as "MLH can be estimated by the measurement of variance of the mechanical turbulence, of the temperature enabling convection or of the substance content in the low troposphere."**(P18L40)**

5. L59: Here ceilometers should be explicitly mentioned. Lidars are comparably rare instruments, and only a few can be operated unattendedly and fully automatic (MPL is one of them, so often called a ceilometer or an "automated low power lidar, ALC"). Ceilometers are available as networks, e.g. in Europe. I don't know the situation in China but this option should be mentioned. It has been demonstrated that most ceilometers are capably to determine the MLH.

**A5:** The comment is added as "With recent upgrades of the hardware, ceilometer, an known as automated low power lidar, or automated lidars and ceilometers, has been demonstrated that be capably to determine the MLH." **(P19L68)**

6. L66: "only in the morning...". This statement is not true for all places in the world.

**A6:** Yes, the statement is not all true, taking into consideration that additional RS may be launched in other time. 08:00 LT and20:00 LT is the universal launch time (Collaud et al., 2014; Guo et al., 2016; Su et al., 2019). The sentence has been rephrased as "The meteorological radiosondes usually acquire the MLH in the morning (08:00 LT) and at night (20:00 LT), when the diurnal cycle of ML combined with stable and convective PBL cannot be well characterized."**(P19L75)**

7. L68: "the low vertical resolution": Most of the ceilometers (and lidars) provide a 15 m resolution that is fully sufficient to determine the CBL. In the framework of Ceilinex2015 we have compared instruments from Vaisala, Lufft, and Campbell. In particular for Lufft CHM15k and Vaisala CL51 no problems appeared (the corresponding AMT-paper was mainly on water vapor absorption corrections).

**A7:** The description is removed.

8. L70: "low SBL height is not evaluated". I assume this is due to the overlap problem. The reader is not aware of this at this point of the manuscript. So a short explanation should be added or the sentence should be moved to the next section.

**A8:** It is moved the next section. **(P21L138)**

9. L85: Here the period of the measurements should be explicitly mentioned, and the time resolution, and the gaps in the measurement schedule if any. The coordinates of the location should be given with more decimal places.

**A9:** The corresponding content has been added "The period of the measurements is from 2013 to 2018, nearly six years. Except for the MLH data from lidar of 2013 mainly existing in winter and spring, the measurement of 2014-2018 are all annual continued observations."**(P20L110)** "CE370 can detect a long range profile up to 30 km every 1 second. For the enhancement of signal noise ratio, 60 profiles are averaged to restore as one with the time resolution of 1 minute."**(P20L99)** "the observation site (116.379° E, 40.005° N) in Beijing city" **(P20L95)**

10. L89: "correction of overlap". As this is an essential point, it would be nice to read a few detail, at least the height of the lowest "useful" range bin should be given here (actually it is mentioned later, sometimes called "gate", sometimes "range").

**A10:** The descriptions of overlap is present in the revised manuscript as "Due to the design of the lidar, the received view close to the ground does not completely coincide with transmitted view. There exist a detection blind area of lidar and a geometric overlap factor is used to correct the mismatch of field of view."**(P20L104)** and "Due to the limitation of algorithm and insufficient lidar overlap, the minimum range of the MLH calculation is on the order of 250 m."**(P22L146)**

11. L94: To my knowledge Baars et al. (2008) were (one of) the first who applied the Haar-wavelet technique to continuous lidar measurements (de Haij et al. used the very old LD-40 with a limited vertical measurement range) including a detailed sensitivity analysis. This paper could be cited as well.

**A11**: It has been added. **(P20L114)**

12. L99: When mentioning "attenuated backscattering profile f(z)" here, this quantity should be defined previously. But in L89 only RCS and "then logarithm calculation" ($\ln(\beta'_{532})$) are mentioned, but not f(z).

**A12**: It is added as "RCS is also expressed as f (z), with z the measurement height." **(P20L107)**

13. L103: The interpretation of the wavelet approach may have some pitfalls: to associate the largest maximum of Wf to the MLH (CBL) is obvious (though exceptions may occur), however, the interpretation of the first local maximum is critical (MLH'). Often the aerosols are structured and several internal layers appear making the allocation of a local maximum to an atmospheric feature very difficult. In the last years a lot of research has been devoted to estimate MLH resulting in a number of papers that should be mentioned here, e.g. Kotthaus et al., Geiß et al., Morille et al., de Bruine et al., Poltera et al. and many more.

**A13:** It is added as "Actually, every local maximum corresponds an aerosol layer and several internal layers appear making the allocation of a local maximum to an atmospheric feature very difficult (Morille et al., 2008; Geiß et al., 2017; Poltera et al., 2017; Kotthaus et al., 2018)."**(P21L127)** And "However, the interpretation of the first local maximum (MLH') is critical. To form a diurnal cycle of MLH from these several layers, a geodesic approach was applied to pathfinderTURB (Poltera et al., 2017), while COBOLT (Geiß et al., 2017) uses a time–height-tracking approach with moving windows. Nevertheless, these method all are based on the

selection of the lowest detected aerosol. The height of the lowest detected aerosol layer was regarded as the daytime MLH and the nocturnal stable boundary layer, respectively, as reported by Mues et al. (2017) and Kotthaus et al. (2018). Su et al. (2019) developed a DTDS algorithm, started with the lowest point and tracked depending time and stability, but the nocturnal MLH is not evaluated. Detection of nocturnal boundary-layer heights, in contrast to the residual layer, is a major challenge (Haeffelin et al., 2012; Lotteraner and Piringer, 2016; de Bruine et al., 2017). SBL seems to be more relevant for the accumulation of pollutants close to the surface than the RL in the evening and early morning. Thus, one of the objective of this study is to investigate the usefulness of $MLH_L$' from CE-370 to capture the SBL height over Beijing."**(P21L132)**

14. L113: It is stated that the temporal resolution of the MLH is one hour. In Figs. 1–4 the resolution seems to be better. Is this a inconsistency?

**A14:** You are right. The resolution of Figs. 1-4 is one minute, according to the primary MLH_lidar result. The description of "For the enhancement of signal noise ratio, 60 profiles are averaged to restore as one thus with the time resolution of 1 minute." **(P20L100)**

15. L114: "eliminating ...false value and peak value": Please give a short hint, what is meant. L133: The description of the PMRS-model is quite relevant, so I suggest to move S2 to the main text. In particular the uncertainty of PM2.5 depending on the uncertainty of the MLH should be highlighted as the MLH and its (large) uncertainty is the main outcome of their study. Fig. 10 reveals extreme differences in case of the RS-retrieval.

**A15:** Due to incomplete screen of cloud and rainfall, it can be misjudged as the MLH, leading to some false and very large value, which should be eliminated. Now the sentence has been rephrased by "unrealistic outliers are deleted". **(P22L145)**

RMRS-model is moved to the main text. **(P23L177)**

"The correlations at 12, 13, 14, and 15 LST were 0.894, 0.922, 0.927, and 0.900, respectively. The higher accuracy may be due to the completed mixing of the aerosol at noon and the vertical distribution of the aerosol tend to be uniform. The correlation between 8, 9 and 17 LST is less than 0.8, and the relatively poor accuracy is related to the complex boundary layer structure in the morning and evening. It is difficult to achieve fully mixing of the aerosol in the stable boundary layer or the residual layer. The daily variation of calculated surface pollutant accuracy using MLH retrieval by lidar vary with the daily variation of aerosol mixing uniformity at different times during the daytime."**(P30L386)**

The calculated PM2.5_RS from $MLH_{RS}$ and in-situ measurement shows great discrepancy, as the reason that "above $MLH_{RS}$ there still exist a large amount of aerosol. The discrepancy makes sense using the method with the determinate total amount of pollutant of the column atmosphere. The gap may be narrowed if the total emission from surface is used."**(P30L378)**

16. L139: What type of sunphotometer is used? Give a few comments on these measurements.

**A16:** The comment is added as "All the parameter is observed by the instruments employed in the same observatory of lidar. The optical parameters of the column aerosols (AOD and FMF.) are obtained by a sky-sun photometer (CE318-DP, CIMEL, France), which is affiliated with the Aerosol RObotic NETwork (AERONET) (Holben et al., 1998; Dubovik, 2000). Measurements are automatically scheduled with direct sun irradiance measurements each of about 15 min and angular sky radiance scanning of about 1 h each (Li et al., 2015; Che et al., 2014; Wang et al., 2019). Atmospheric meteorological data (relative humidity-RH, wind speed-WS, wind direction-WD, etc.) are obtained by automatic meteorological monitoring station (BLJW-4). $PM_{2.5}$ mass concentration is obtained by $PM_{2.5}$ monitor (BAM-1020, MetOne, USA), which shows good agreement with the measurement of national monitoring network near the observatory. All the data is quality controlled and calculated as one hour averaged and the measurement period is from 2014 to 2018."**(P23L184)**

17. Section 3.1: "case study" is mentioned in the caption, but it is not very clear, what this is. Four case studies each covering a period of 3 days corresponding to Figs.1-4?

**A17:** "case study" in the caption is removed in the revised manuscript.**(P24L196)**

18. Figs. 1–4: The triangles cover a vertical range of 300 m or so. What part of the symbol indicates the RS-retrieval? The top, the center, the bottom? In many cases the MLH seems to be zero.

**A18:** The top of the triangle indicats the RS-retrieval.

19. L151: "the aerosol layer height keep...": This sentence should be rephrased.

**A19:** It has been rephrased.**(P24L202)**

20. L152: "MLH is always higher than MLH_RS,...": this seems to be in contradiction to the findings in L204. Please check all conclusions carefully.

**A20:** The sentence of "MLH is always higher than MLH_RS,..." is removed, while "MLH_RS tends to be larger than MLH" is rephrased as "MLH_RS tends to be larger than MLH in the afternoon". **(P26L263)** Actually, we want to express that MLH tend to be higher than MLH_RS in the evening and early morning, while MLH is lower than MLH_RS in the afternoon.

21. L175: "...RS is its very good precision". This should be explained. Why is the method conceptually superior to a ceilometer/lidar retrieval? The criteria involved for estimating the MLH from RS or lidars both have their "free parameters" (thresholds).

**A21:** The paper focus the measurement of lidar. Indeed, it is improper to emphasize the precision of RS. Yet, from my perspective ,in view of the definition of MLH inferring "convection or mechanical turbulence" (COST action 710 – Final report, 1998), RS measurement is likely to close to the two parameter, "due to their ability to characterize the thermodynamic and dynamic states of the boundary layer".**(P22L154)** And, a lot of study has been done to evaluate the precision of remote sensing measurement, with the comparison with RS (Wiegner et al., 2006; Milroy et al.,

2012; Sawyer and Li, 2013; Cimini et al., 2013; Tang et al, 2016; Singh et al., 2016; Mues et al., 2017; Su et al, 2019).**(P19L71)** And, in section 2.3, there is the statement of "In most cases, the exact threshold value has only a small impact on the PBL height due to the large slope of $Ri_b$ in this interval (Collaud et al., 2014)."**(P23L169)**

Nevertheless, the sentence has been rephrase as "The temporal resolution (usually two or three measurements per day) of PBL detection by RS is not able to provide the mixing layer height diurnal cycle, no matter its good precision."**(P31L400)**

22. L190: The histograms should be explained in much more detail to avoid confusion. The reader might expect that the columns (e.g. for winter) add up to 100 % (for the lidar and the RS retrieval). However, it seems that the total column is the annual relative frequency and the different colors indicate the contribution of each season to the total. If so, the seasonal distribution should be discussed as well. The overall agreement between the two data sets is actually low – neither the absolute values nor the shape of the distribution agree. Moreover, as stated by the authors, in 35 % of the cases no intercomparison is possible due to the overlap problem. This mainly occurs in spring – any idea why?

**A22:** Yes, It is added that "the total column is the annual relative frequency and the different colors indicate the contribution of each season to the total."**(P25L238)** The description of Fig. 5 is rephrased that "Fig. 5. Comparison of frequency distribution of all MLH (2013-2018) retrieved from lidar and radiosonde with the supplementary information of seasonal variation. Noted that for presenting the detail distribution, $MLH_L$ adds up to 20%, while $MLH_{RS}$ add up to 45%."**(P43L661)**

Actually, the seasonal distribution was discussed in the manuscript, as "As to the seasonal variation of both lidar and RS measurement at 0800 LST, the frequency of larger $MLH_L$ value in summer is minimal, indicating summer MLH is lower than other season. As for radiosonde, $MLH_L$ lower than 0.25 km mostly distributes in winter, with the rate of around 15% for both 0800 and 2000 LST, and the frequency decreases rapidly when $MLH_L$ gets larger than 0.25 km."**(P25L249)**

It is added that "This lower values mainly occurs in winter and autumn, when it tends to present lower MLH (Tang et al., 2016)."**(P25L241)**

23. Figs. 7 and 8: to compare these two figures (basically it is the same information, however, in Fig. 7 the annual mean and in Fig. 8 the seasonal means are shown?) relevant minimum/maximum values of the MLH should explicitly be given in the text. Then, it can be seen if the numbers are consistent. What is the "shaded area" in Fig. 7: from 550 m to 2000 m in case of MLH at 0000 hours? What are the consequences of such a large range for the significance of differences (in the course of the day, for inter-annual changes)?

**A23:** Besides the annual mean diurnal cycle, Fig. 7 also compared the mean MLH with MLH and MLH_RS, revealing that MLH shows overall good CBL height and MLH' generally cannot indicate SBL well. In the revised manuscript, Fig. 7 has been remove to the section of comaparisong of different MLH approach. The seasonal and annual variation is presented mainly based on MLH.

The content and table is added. "The maximum in summer is 1.526 ± 0.581 km, and the maximum in autumn is 1.445 ± 0.837 km. From the all-day average of the four seasons, the averages in spring, summer, autumn and winter are 1.409 km, 1.261 km, 1.297 km and 1.228 km, respectively."**(P27L310)**

Table 1 Statistics of boundary layer height seasonal change **(P50L700)**

| MLH/km | Spring | summer | autumn | winter |
|---|---|---|---|---|
| mean | 1.409 | 1.261 | 1.297 | 1.228 |
| maximum | 1.647 | 1.526 | 1.445 | 1.404 |
| minimum | 1.126 | 0.932 | 1.117 | 1.098 |

The "shaded area" in Fig. 7 indicates the standard deviation of MLH and MLH'. In statistics, mean ± standard deviation is commonly used to indicate the average and degree of dispersion of a set of data, referred Su et al. (2019) and Geiß et al., (2017). The inter-annual changes is analyzed based on the mean result, even if the data dispersion is large, it still has statistical significance.

24. L228: According to the authors MLH' could be the residual layer or the stable layer (or any internal layer). So, the implications for the dispersion of pollutants are hard to infer. The benefit of MLH' should be clearly described in the paper (see comment above, and the conclusions of the manuscript).

**A24:** It is added in the conclusions that "$MLH_L$' have the potential to describe the stable layer height at night sometime, even though the capability is limited due to the high incomplete overlap of CE-370. The stable layer height detected by $MLH_L$' in the nighttime is the layer in which ground-emitted atmospheric pollutants are trapped, it contributes to the assessment of the surface pollutant concentration when there is emission in the nocturnal time using the numerical models."**(P31L403)**

25. L236: The authors explain the "valley" in the diurnal cycle from the domination of the developing CBL over the RL (in terms of the signal gradient) after sunrise. In many publications the complete diurnal cycle is considered as the combination of the SL and the CBL. Then, a much smoother curve can be found. Moreover, the SL seems to be more relevant for the accumulation of pollutants close to the surface than the RL.

**A25:** The application of height of SL and RL may be different, as discussed in the conclusion section that "The stable layer height detected by $MLH_L$' in the nighttime is the layer in which ground-emitted atmospheric pollutants are trapped, it contributes to the assessment of the surface pollutant concentration when there is emission in the nocturnal time using the numerical models."**(P27L291)** "Whilst the residual layer height corresponding to trapped atmospheric constituents discharged some hours before, which can be employed to convert column-mean optical depths into near-surface air quality information from remote sensing."**(P27L301)**

26. L259: "...into near-surface air quality information". It has been shown by Geiß et al. (2017) that the near-surface air quality does not only depend on the MLH (see also comment L33, and manuscript L275).

**A26**: Please see the response to the comment 3.

27. Fig. 9: Are the inter-annual differences of the annual cycles significant in view of the very large uncertainty ranges (see also comment above)?

**A27:** The inter-annual changes is analyzed based on the mean result, even if the data dispersion is large, it still has statistical significance.

28. L262: A short description of the in-situ measurements should be added: where have they been made, what is their temporal resolution/coverage, and their accuracy. Is one site or an average over many sites considered? The differences between MLH_RS and the in-situ seem to be indeed too large for any air quality application.

**A28:** It is added that "All the parameter is observed by the instruments employed in the same observatory of lidar. The optical parameters of the column aerosols (AOD and FMF.) are obtained by sky-sun photometer (CE318-DP, CIMEL, France), which is affiliated with the Aerosol RObotic NETwork (AERONET) (Holben et al., 1998; Dubovik, 2000). Measurements are automatically scheduled with direct sun irradiance measurements each of about 15 min and angular sky radiance scanning of about 1 h each (Li et al., 2015; Che et al., 2014; Wang et al., 2019). Atmospheric meteorological data (relative humidity-RH, wind speed-WS, wind direction-WD, etc.) are obtained by automatic meteorological monitoring station (BLJW-4). PM2.5 mass concentration is obtained by PM2.5 monitor (BAM-1020, MetOne, USA), which shows good agreement with the measurement of national monitoring network near the observatory. All the data is quality controlled and calculated as one hour averaged and the measurement period is from 2014 to 2018." **(P23L184)**

Like to the response of comment 15, the calculated $PM2.5\_RS$ from $MLH_{RS}$ and in-situ measurement shows great discrepancy, as the reason that "above $MLH_{RS}$ there still exist a large amount of aerosol. The discrepancy makes sense using the method with the determinate total amount of pollutant of the column atmosphere. The gap may be narrow if the total emission from surface is used." **(P30L378)**

29. L297: Last sentence: the authors should be aware that many sophisticated methods to retrieve the diurnal cycle has been published recently. They should be cited (many references in the manuscript are quite old), see suggestions.

**A29:** The reference of "Wiegner et al., 2006; de Bruine et al., 2017; Morille et al., 2017; Kotthaus et al., 2018" is cited.**(P32L428)**

**Reply to Anonymous Referee #2:**

**General comments:**

This manuscript reports climatological values of mixing layer height (MLH) over Beijing for from Lidar measurements, their evaluation against radiosonde based MLH calculation and argues that MLH values derived from Lidar is better than that from RS for estimation of PM2.5 at surface. The manuscript needs a lot of improvement before publication. Please see comments and suggestions below:

1. The first part of the results and discussion, where the comparison of the various approaches to estimate MLH is presented is not satisfactory. First, the English language is poorly written which makes it hard to understand what is being conveyed. Also, the text corresponding to Figures 2-4 lacks any discussion of the features seen in them. More detailed discussion and language improvement is needed.

**A1**: This part has been rephrased, please see the revised manuscript.

2. Fig 3d: why is MLH_RS detected at 0.6 km? it should be 2km as seen in Fig 3c.

**A2:** The vertical profile present is actually indicated by "the white edge triangle in the upper picture", also the measurement time can be seen on the top of Fig. 3c (20170606-14) and Fig. 3d (20170606-20). Fig. 3d show $MLH_{RS}$ (0.6 km) at 2000 LST, while Fig.3c indicates (2 km) at 1400 LST. Due to the different time, $MLH_{RS}$ varies.

3. Actually, the whole context of the first 4 figures is not understood. Why are these days shown here? why not any other day? Is it meant to show seasonal variability i.e one for one season? If the figures are introduced to show various types of differences between the 3 methods, then the discussion show be organized in that manner and the inter comparison figure 5 should be discussed in context of features seen in these figures 1-4. Better organization is needed.

**A3:** Fig. 1-Fig. 4 present as case study is aimed to show the evolution of the MLH and comparisons of lidar measurement ($MLH_L$ and $MLH_L'$) and RS measurement ($MLH_{RS}$). The four cases is almost all cloud free to present continuous retrievals. Fig.1 shows obvious separated layers in the evening (spring without 1400LST measurement). Fig.2 shows low aerosol load, in which condition lidar can still capture the small gradient (summer with 1400 LST data). Fig. 3 relative high aerosol load and MLHL and MLHL' show obvious difference in the afternoon (summer with 1400 LST data).Fig. 4 keep stable though the whole day. The information delivered of the figure are discussed in the following context in the revised manuscript.

4. Please give more details about the cloud screen/flag used (Line 111). How was cloud detected, at what resolution etc?

**A4:** The content is added that "a threshold is selected to distinguish between clouds and aerosol layers."**(P22L144)** and the lidar "can detect a long range profile up to 30 km every 1 second. For

the enhancement of signal noise ratio, 60 profiles are averaged to restore as one with the time resolution of 1 minute."**(P20L99)**

5. Fig 1a, 2a,3a,4a all are stretched, the fonts are of different sizes compared to other panels and should be made consistent. Figure 1b have an undefined variable MLH2 in the figure label?

**A5:** The layout of the sub figure is based on requirement of clarity, expressiveness and artistry. In our view of Fig. 1a-4a, the information conveyed changes nothing. In Figure 1b, MLH2 has been revised to $MLH_L'$, which indicates the first local maximum of lidar MLH.

6. Evaluation of MLH_lidar is essential for second half of this manuscript. Hence, more detailed comparison and discussion is necessary. scatter plots in supplementary should be presented in main and discussed in detail. Moreover, as seasonal variability is significant on MLH, the comparison in Figure 6 should include and discuss scatter plots for all the seasons, separately.

**A6:** The comparisons of MLH from lidar ($MLH_L$ and $MLH_L'$) and $MLH_{RS}$ is the one of objective of the study. From Fig. 5, we can see that about 35% SBL is not at the lidar detection range for 0800 and 2000 LST. It can be concluded that the agreement of MLH from lidar ($MLH_L$ and $MLH_L'$) and $MLH_{RS}$ is poor, even though the scatter plots is not shown. The comparisons between $MLH_L'$ and $MLH_{RS}$ in the afternoon can be seen from the case study and the diurnal cycle, which indicates $MLH_L'$ is much lower than $MLH_{RS}$. Please see the revised manuscript. Actually, the data of radiosonde is only available in summer, so the comparisons of the other seasons cannot be conducted.

7. From Fig 1-5, MLH from Lidar is termed as MLH but in Fig 6, it is termed as MLH_Lidar. Please be consistent.

**A7:** All the biggest local maximum MLH of lidar is represented by $MLH_L$, while the first local maximum is indicated by $MLH_L'$ and radiosonde result by $MLH_{RS}$. They are consistent in through the revised manuscript.

8. Fig 9: Is the MLH from RS also showing yearly variability similar to the MLH derived from Lidar? Please include the same from RS also in this figure and discuss.

**A8:** From 2014 to 2018, $MLH_{RS}$ at 0800 LST is 0.402, 0.412, 0.453, 0.444, 0.451 km, respectively. $MLH_{RS}$ at 2000 LST is 0.445, 0.501, 0.512, 0.515, 0.480 km, respectively. Both of the two measurement time show increasing trend. Due the availability of RS measurement is only in summer, it cannot compare with the annual mean value of lidar. We can see the annual variation of RS is not exactly the same as lidar. Actually, in the revised manuscript, the compassion of lidar and RS measurement is discussed it the prior section, and in the section, we focus to present the annual variation of diunal cycle of MLH from lidar, while the RS measurements is just time points result.

9. In Figure 10d shows that correlations are highest when RMSE is also highest, this is non intuitive, please describe this feature. Also, the comparison in Figure 10 should be presented separately for each season as MLH has strong seasonal variations.

**A9:** The comment is added in the revised manuscript that "The smaller RMSE is related to the limited samples." **(P30L390)** And "larger RMSE is associated to the larger amount of samples into statistic."**(P30L381)** In the revised manuscript, there is already a lot of content. The seasonal variations of calculated PM2.5 and in-situ can be shown in the next paper.

10. Throughout the manuscript standard deviation and RMSE is used interchangeable, which should be corrected. They are not same.

**A10:** Thank you for reminding, and the standard deviation has been revised to RMSE.

11. Please check carefully for the typos in the manuscript. The title itself has one "airpollution"

**A11:** Thank you for reminding and I have noticed the typos. Hope the staff member of ACP could help to correct it. Actually, in my manuscript, it is right "air polllution".

[revised manuscript text omitted]